# Interpretable Hierarchical Concept Reasoning through Graph Learning

## Abstract

Concept-Based Models (CBMs) are a class of deep learning models that provide interpretability by explaining predictions through high-level concepts. These models first predict concepts and then use them to perform a downstream task. However, current CBMs offer interpretability only for the final task prediction, while the concept predictions themselves are typically made via black-box neural networks. To address this limitation, we propose Hierarchical Concept Memory Reasoner (H-CMR), a new CBM that provides interpretability for both concept and task predictions. H-CMR models relationships between concepts using a learned directed acyclic graph, where edges represent logic rules that define concepts in terms of other concepts. During inference, H-CMR employs a neural attention mechanism to select a subset of these rules, which are then applied hierarchically to predict all concepts and the final task. Experimental results demonstrate that H-CMR matches state-of-the-art performance while enabling strong human interaction through concept and model interventions. The former can significantly improve accuracy at inference time, while the latter can enhance data efficiency during training when background knowledge is available.

## 1 Introduction

Concept-Based models (CBMs) have introduced a significant advancement in deep learning (DL) by making models explainable-by-design (Koh et al., 2020; Alvarez Melis & Jaakkola, 2018; Chen et al., 2020; Espinosa Zarlenga et al., 2022; Mahinpei et al., 2021; Debot et al., 2024; Barbiero et al., 2023; Poeta et al., 2023; Dominici et al., 2024; Vandenhirtz et al., 2024; Espinosa Zarlenga et al., 2023; Havasi et al., 2022). These models integrate high-level, human-interpretable concepts directly into DL architectures, bridging the gap between black-box neural networks and transparent decision-making. One of the most well-known CBMs is the Concept Bottleneck Model (CBNM) (Koh et al., 2020), which first maps an input (e.g. an image) to a set of human-understandable concepts (e.g. "pedestrian present," "danger in front") using a neural network and then maps these concepts to a downstream task (e.g. "press brakes") via a linear layer. The predicted concepts serve as an interpretable explanation for the final decision (e.g. "press brakes because there is a danger in front"). Considerable research has focused on ensuring CBMs achieve task accuracy comparable to black-box models. Some CBMs (Espinosa Zarlenga et al., 2022; Mahinpei et al., 2021; Debot et al., 2024; Barbiero et al., 2023) are even known to be *universal classifiers* (Hornik et al., 1989); they match the expressivity of neural networks for classification tasks, regardless of the chosen set of concepts.

While CBMs enhance interpretability at the task level, their concept predictions remain opaque, functioning as a black-box process. Most CBMs model the concepts as conditionally independent given the input (e.g. Figure 1a), meaning any dependencies between them must be learned in an opaque way by the underlying neural network (Koh et al., 2020; Alvarez Melis & Jaakkola, 2018; Chen et al., 2020; Espinosa Zarlenga et al., 2022; Mahinpei et al., 2021; Debot et al., 2024; Barbiero et al., 2023). In contrast, some existing approaches attempt to capture relationships between concepts in a structured way (Dominici et al., 2024) (e.g. Figure 1b). However, the current approaches do not really provide interpretability: while they reveal which concept predictions influence others, they do not explain *how* these influences occur. For instance, one can determine that 'pedestrian present' has some influence on 'danger in front,' but not the exact nature of this influence.

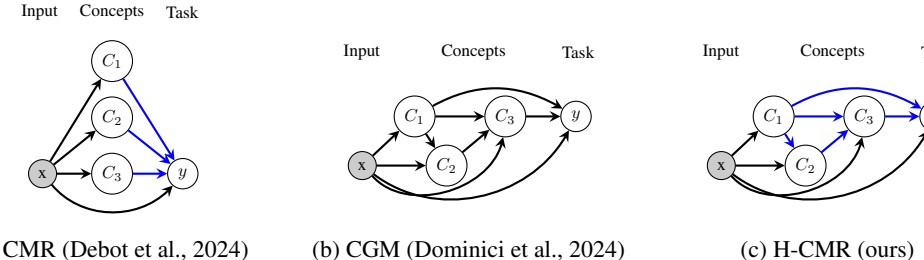

(a) CMR (Debot et al., 2024)     (b) CGM (Dominici et al., 2024)     (c) H-CMR (ours)

Figure 1: Comparison of example CBMs. Blue and black edges are interpretable and black-box operations, respectively. (a) Some approaches model conditionally independent concepts with interpretable task inference. (b) Others learn a hierarchy of concepts with black-box task inference. (c) Only our approach learns a hierarchy with interpretable inference for both concepts and task.

In this paper, we introduce Hierarchical Concept Memory Reasoner (H-CMR) (Figure 1c), the first CBM that is both a universal classifier and provides interpretability at both the concept and task levels. H-CMR learns a *directed acyclic graph* (DAG) over concepts and tasks, which it leverages for inference. H-CMR employs *neural rule generators* that produce symbolic logic rules defining how concepts and tasks should be predicted based on their parent concepts. These generators function as an attention mechanism between the input and a jointly learned memory of logic rules, selecting the most relevant rules for each individual prediction. Once the rules are selected for a specific input, the remaining inference is entirely interpretable for both concepts and tasks, as it follows straightforward logical reasoning. This means the human can inspect how parent concepts exactly contribute to their children. By combining graph learning, rule learning, and neural attention, H-CMR provides a more transparent and structured framework for both concept predictions and downstream decision-making.

Our experiments demonstrate that H-CMR achieves state-of-the-art accuracy while uniquely offering interpretability for both concept and task prediction. This interpretability enables strong human-AI interaction, particularly through interventions. First, humans can do *concept interventions* at inference time, correcting mispredicted concepts. Unlike in typical CBMs, which model concepts as conditionally independent, these interventions are highly effective: correcting one concept can propagate to its dependent concepts, potentially cascading through multiple levels. Second, humans can do *model interventions* at training time, modifying the graph and rules that are being learned, allowing the human to shape (parts of) the model. We show that this enables the integration of prior domain knowledge, improving data efficiency and allowing H-CMR to perform in low-data regimes.

We want to stress that H-CMR does not attempt to learn the causal dependencies between concepts *in the data*. Instead, H-CMR reveals its internal reasoning process: which concepts does the model use for predicting other concepts (also known as *causal transparency* (Dominici et al., 2024)), and how (*interpretability*).

## 2 MODEL

We introduce Hierarchical Concept Memory Reasoner (H-CMR), the first CBM that is a universal binary classifier providing interpretability for both concept and task predictions. We considered three main desiderata when designing H-CMR (for more details, see Section 5): **interpretability**, the ability for humans to understand how concepts and tasks are predicted using each other; **intervenability**, the ability for humans to meaningfully interact with the model; **expressivity**, a requirement for the model to be able to achieve similar levels of accuracy as black-box models irrespective of the employed set of concepts. For simplicity, we only consider concepts in the remaining sections, omitting the task. The task can be treated similar as the concepts, or separately like in most CBMs (see Appendix A).

### 2.1 HIGH-LEVEL OVERVIEW

From a high-level perspective, H-CMR consists of three main components (Figure 2): a *rule memory*, an *encoder*, and a *decoder*. The encoder predicts a small number of concepts and an embedding, and the decoder selects from the memory a set of logic rules to hierarchically predict all other concepts.

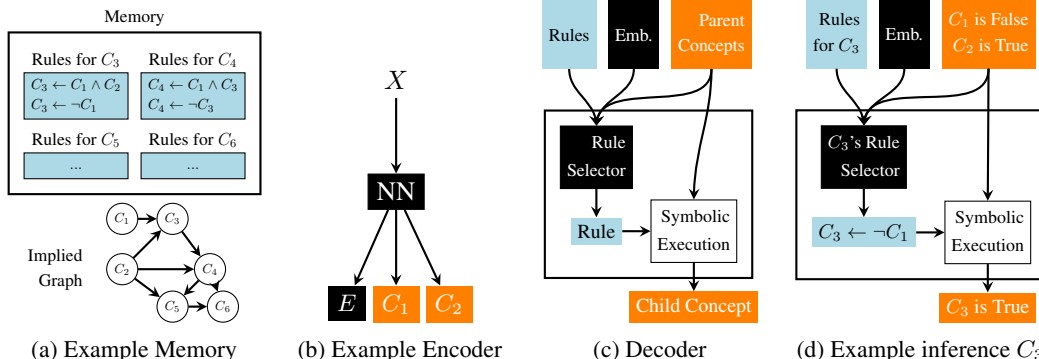

(a) Example Memory  (b) Example Encoder  (c) Decoder  (d) Example inference $C_3$

Figure 2: High-level overview of the different components of H-CMR. (a) The memory is compartmentalized per concept and implies a DAG over the concepts. (b) The encoder predicts source concepts and an embedding. (c) The decoder infers each non-source concept from its parent concepts, the embedding and its rules, by selecting a rule (using a neural network) and then symbolically executing that rule using the parent concepts. (d) Example of the decoder predicting $C_3$.

As standard in CBMs, concepts are predefined and come with the dataset, and they are supervised, with labels in the dataset or extracted using Vision-Language Models (Oikarinen et al., 2023a).

**Memory.** The memory is compartmentalized per concept. For each concept, it stores a set of (learned) logic rules that define that concept in terms of other concepts, e.g. $C_3 \leftarrow C_1 \wedge C_2$, or $C_3 \leftarrow \neg C_1$. This memory implicitly defines a *directed acyclic graph* (DAG) over the concepts: if a concept $C_i$ appears in at least one rule defining another concept $C_j$, then $C_i$ is a parent of $C_j$. For some concepts, all associated rules will be "empty" (e.g. $C_i \leftarrow .$), meaning they have no parents. These are the *source concepts* of the DAG, which are predicted directly by the encoder rather than inferred via rules.

**Encoder.** The encoder is a neural network which maps the input to the source concepts and a latent embedding. Thus, source concepts are directly predicted from the input in the same black-box fashion done by standard CBMs.[1] The latent embedding captures additional contextual information from the input that may not be captured by the concepts. This preserves the concept-prediction expressivity of other CBMs, and the task-prediction expressivity of black-box neural networks (see Section 5).

**Decoder.** The decoder is used to hierarchically perform inference over the concept DAG. At each step, it predicts a concept using its parent concepts, the latent embedding, and its rules in the memory. It leverages a neural attention mechanism to select the most relevant rule for the current prediction, based on the parent concepts and the latent embedding. This rule is then symbolically executed on the parent concepts to produce the concept prediction.

This approach is designed to handle settings where (i) no graph over concepts is available and must therefore be learned from data, (ii) rules defining concepts in terms of others are unknown and must be learned, and (iii) the available concepts alone are *insufficient* for perfect prediction, requiring additional contextual information, here exploited through the rule selection and the latent embedding. While the learned rules may be noisy in settings where concepts are insufficient (i.e. directly applying all rules may not lead to correct predictions in every case), this is not a problem due to the selection mechanism, as only the *selected rule* is required to yield the correct prediction.

## 2.2 PARAMETRIZATION

In this section, we go into more detail on how the individual components are parametrized. We refer to Appendix C for H-CMR's probabilistic graphical model, and Appendix B for more details regarding the neural network architectures. We explain how these components are used to do inference in Section 3.

---

[1] While standard CBMs do this for all concepts, H-CMR only does this for source concepts.

### 2.2.1 ENCODER

The encoder directly predicts each source concept $C_i$ and the embedding $E$ from the input $x$:

$$p(C_i = 1 \mid \hat{x}) = f_i(\hat{x}), \quad \hat{e} = g(\hat{x}) \tag{1}$$

where each $f_i$ and $g$ are neural networks, with the former parametrizing Bernoulli distributions.[2] This deterministic modelling of the embedding $E$ corresponds to a delta distribution.

### 2.2.2 DECODER

The decoder infers non-source concepts from their parent concepts, the latent embedding and the rules in the memory, and operates in two steps. First, the parent concepts and the embedding are used to select a logic rule from the set of rules for that concept. Second, this rule is evaluated on the parent concept nodes' values to produce an interpretable prediction. More details on this memory and the representation and evaluation of rules are given in Section 2.2.3.

The selection of a rule for a concept $C_i$ is modelled as a categorical random variable $S_i$ with one value per rule for $C_i$. For instance, if there are three rules for $C_1$ and the predicted categorical distribution for $S_1$ is $(0.8, 0.2, 0.0)$, then this means that the first rule in the memory is selected with 80% probability, the second rule with 20%, and the third with 0%. The logits of this distribution are parametrized by a neural network that takes the parent concepts and latent embedding as input. For each non-source concept $C_i$, the concept prediction is:

$$p(C_i = 1 \mid \hat{e}, \hat{c}_{parents(i)}, \hat{r}_i) = \sum_{k=1}^{n_R} \underbrace{p(S_i = k \mid \hat{e}, \hat{c}_{parents(i)})}_{\substack{\text{neural selection of rule } k \\ \text{using parent concepts + emb.}}} \cdot \underbrace{l(\hat{c}_{parents(i)}, \hat{r}_{i,k})}_{\substack{\text{evaluation of concept } i\text{'s rule } k \\ \text{using parent concepts}}} \tag{2}$$

with $\hat{e}$ the latent embedding, $n_R$ the number of rules for each concept, $\hat{r}_i$ the set of rules for this concept, $S_i$ the predicted categorical distribution over these rules, and $l(\hat{c}, \hat{r}_{i,k})$ the symbolic execution of rule $k$ of concept $i$ using concepts $\hat{c}$ (see Section 2.2.3). Intuitively, all the learned rules for $C_i$ contribute to its prediction, each weighted by the rule probability according to the neural selection.

### 2.2.3 MEMORY, RULE REPRESENTATION AND RULE EVALUATION

For each concept, H-CMR learns $n_R$ rules in its memory, with $n_R$ a hyperparameter. The memory and the representation of rules resemble the approach of Debot et al. (2024).[3] For each concept $C_i$, the memory contains $n_R$ embeddings, each acting as a latent representation of a rule. These embeddings are decoded using a neural network into symbolic representations of logic rules, enabling symbolic inference. We consider rule bodies that are conjunctions of concepts or their negations, e.g. $C_3 \leftarrow C_0 \wedge \neg C_1$ (read "if $C_0$ is true and $C_1$ is false, then $C_3$ is true").

A rule is represented as an assignment to a categorical variable over all possible rules. Explicitly defining this distribution would be intractable, as there are an exponential number of possible rules. Instead, we factorize this variable into $n_C$ independent categorical variables $R_i$, each with 3 possible values corresponding to the *role* of a concept in the rule. For instance, in $C_3 \leftarrow C_0 \wedge \neg C_1$, we say $C_0$ plays a *positive* ($R_0 = P$) role, $C_1$ plays a *negative* ($R_1 = N$) role, and $C_2$ is *irrelevant* ($R_2 = I$).

Evaluating a rule on concept predictions follows the standard semantics of the logical connectives. Using our representation of a rule, this becomes:

$$l(\hat{c}, \hat{r}_{i,k}) = \prod_{j=1}^{n_C} (\mathbb{1}[\hat{r}_{i,k,j} = P] \cdot \mathbb{1}[\hat{c}_j = 1] + \mathbb{1}[\hat{r}_{i,k,j} = N] \cdot \mathbb{1}[\hat{c}_j = 0]) \tag{3}$$

where $n_C$ is the number of concepts, $l(\cdot)$ is the logical evaluation of a given rule using the given concepts $\hat{c}$, and $\hat{r}_{i,k,j}$ is the role of concept $j$ in that rule (positive (P), negative (N) or irrelevant (I)).

Decoding each rule embedding into this symbolic representation (i.e. assignments to $n_C$ categorical variables $R_i$) happens in two steps, and is different from Debot et al. (2024). First, a neural network

---

[2]We abbreviate the notation for assignments to random variables, e.g. $\hat{x}$ means $X = \hat{x}$.

[3]Note that their rules define tasks in terms of concepts. Ours also define concepts in terms of each other.



(a) All possible edges allowed by the node priorities ($O$).

(b) Example graph imposed by un-adjusted rules ($R'$).

(c) Example DAG imposed by adjusted rules ($R$).

Figure 3: Example of H-CMR's learned DAG over concepts as defined by its learned rules. The learned node priority vector $O$ (i.c. $O_0 < O_1 < O_2 < O_3$) enforces a topological ordering of the nodes, guaranteeing that the learned graph is a DAG.

maps each rule embedding to the logits for $n_C$ categorical distributions $R'_i$. Then, to ensure that the rules form a DAG, we must prevent cyclic dependencies. For instance, we should not learn conflicting rules such as $C_1 \leftarrow C_0$ and $C_0 \leftarrow C_1$, or $C_1 \leftarrow C_1$. To enforce this constraint, we draw inspiration from Massidda et al. (2023), defining a learnable node priority vector which establishes a topological ordering over concepts: higher-priority concepts are not allowed to appear in the rules of lower-priority concepts, thereby preventing cycles. We achieve this by using the node priorities to modify the categorical distributions $R'_i$, obtaining the to-be-used distributions $R_i$. Specifically, we make any rule that violates the ordering impossible, ensuring its probability is zero. Let $O_i$ be the node priority of concept $i$, then:

$$\forall r \in \{P, N\} : p(R_{i,k,j} = r) = \mathbb{1}[O_j > O_i] \cdot p(R'_{i,k,j} = r) \tag{4}$$

$$p(R_{i,k,j} = I) = \mathbb{1}[O_j > O_i] \cdot p(R'_{i,k,j} = I) + \mathbb{1}[O_j \le O_i] \tag{5}$$

where $\mathbb{1}[\cdot]$ is the indicator function, and $p(R_{i,k,j})$ is the categorical distribution of the role of concept $j$ in rule $k$ for concept $i$ corrected with the node priorities $O$, which are delta distributions. Figure 3 gives a graphical example. The employed rules in the memory are assignments to these random variables, which are used in Equation 3. During training, these assignments are obtained by sampling from this distribution (see Section 4). During inference, we take the most likely roles (see Section 3).

## 3 INFERENCE

For the derivation of the equations below from H-CMR's probabilistic graphical model, we refer to Appendix C. Computing the exact likelihood of a concept corresponds to:

$$p(C_i|\hat{x}) = \sum_{\hat{c}_{parents(i)}} \mathbb{1}[Source_i = 1] \cdot p(C'_i \mid \hat{x}) + \mathbb{1}[Source_i = 0] \cdot p(C''_i \mid \hat{x}, \hat{c}_{parents(i)}, \hat{r}_i) \tag{6}$$

where $\hat{r}_i$ is given by Equation 8, $p(C'_i \mid \cdot)$ by Equation 1 and $p(\hat{C}''_i \mid \cdot)$ by Equation 2. Concepts are source concepts if all of their rules are empty, i.e. for all their rules, each concept is irrelevant: $Source_i = \prod_{k=1}^{n_R} \prod_{j=1}^{n_C} \mathbb{1}[\hat{r}_{i,k,j} = I]$. The sum goes over all possible assignments to the parent concepts. As this would make inference intractable, we instead take an approximation of the Maximum A Posteriori estimate over the concepts by thresholding each individual concept prediction at 50%:[4]

$$p(C_i|\hat{x}) = \mathbb{1}[Source_i = 1] \cdot p(C'_i \mid \hat{x}) + \mathbb{1}[Source_i = 0] \cdot p(C''_i \mid \hat{x}, \hat{c}_{parents(i)}, \hat{r}_i) \tag{7}$$

with $\hat{c}_{parents(i)} = \{\mathbb{1}[p(C_j = 1 \mid \hat{x}) > 0.5] \mid \mathbb{1}[Parent_{ij} = 1]\}$. Note that this thresholding is also beneficial for avoiding the problem of concept leakage in CBMs, which harms interpretability (Marconato et al., 2022). As mentioned earlier, a concept is another concept's parent if it appears in at least one of its rules: $Parent_{ij} = 1 - \prod_{k=1}^{n_R} \mathbb{1}[\hat{r}_{i,k,j} = I]$ (i.e. in not all rules for $C_i$, the role of $C_j$ is irrelevant). Finally, the roles $\hat{r}_i$, which represent logic rules, are the most likely roles:

$$\hat{r}_{i,k,j} = \arg\max_{r \in \{P,N,I\}} p(R_{i,k,j} = r) \tag{8}$$

---

[4]An alternative is to sample the concepts instead.

## 4 LEARNING PROBLEM

During learning, H-CMR is optimized jointly: the encoder (neural network), the decoder (rule selector neural networks) and the memory (node priority vector, rule embeddings and rule decoding neural networks). The training objective follows a standard objective for CBMs, maximizing the likelihood of the concepts. For the derivation of this likelihood and the other equations below using H-CMR's probabilistic graphical model, we refer to Appendix C. Because the concepts are observed during training, this likelihood becomes:

$$\max_{\Omega} \sum_{(\hat{x},\hat{c}) \in \mathcal{D}} \sum_{i=1}^{n_C} \log p(\hat{c}_i \mid \hat{c}, \hat{x}) \tag{9}$$

where we write $\hat{c}_{parents(i)}$ as $\hat{c}$ to keep notation simple. Each individual probability is computed using Equation 1 for source concepts and Equation 2 for other concepts:

$$p(\hat{c}_i|\hat{c},\hat{x}) = \mathbb{E}_{\hat{r} \sim p(R)} \left[ p(\hat{c}_i' \mid \hat{x}) \cdot p(Source_i \mid \hat{r}) + p(\hat{c}_i'' \mid \hat{c}, \hat{e}, \hat{r}) \cdot p(\neg Source_i \mid \hat{r}) \right] \tag{10}$$

$$\text{where} \quad p(Source_i \mid \hat{r}) = \prod_{j=1}^{n_C} \prod_{k=1}^{n_R} \mathbb{1}[R_{i,k,j} = I] \tag{11}$$

with $p(C_i' \mid \hat{x})$ and $\hat{e}$ corresponding to Equation 1, $p(C_i'' \mid \hat{c}, \hat{e}, \hat{r})$ to Equation 2, and $p(R)$ to Equation 4.[5] Note that the designation of source concepts and parent concepts may change during training, as the roles $R$ change. Additionally, to promote learning rules that are *prototypical* of the seen concepts, we employ a form of regularization akin to Debot et al. (2024) (see Appendix B).

**Scalability.** Computing the above likelihood scales $O(n_R \cdot n_C^2)$ at training time with $n_C$ the number of concepts and $n_R$ the number of rules per concept. At inference time, this is the worst-case complexity, depending on the structure of the learned graph.

## 5 EXPRESSIVITY, INTERPRETABILITY AND INTERVENABILITY

### 5.1 EXPRESSIVITY

H-CMR functions as a universal binary classifier for both concept and task prediction. This means it has the same expressivity of a neural network classifier, regardless of the employed concepts. In practice, this translates to high accuracy , irrespective of the chosen concept set.

**Proposition 5.1.** *H-CMR is a universal binary classifier (Hornik et al., 1989) if $n_R \geq 2$, with $n_R$ the number of rules for each concept and task.*

Furthermore, H-CMR's parametrization guarantees that the learned graph over the concepts forms a DAG, and is expressive enough to represent any possible DAG, meaning any possible dependency structure. Let $\Theta$ be the set of all possible parameter values H-CMR can take. For a specific parameter assignment $\theta \in \Theta$, let $\mathcal{G}_\theta$ represent the corresponding concept graph.

**Proposition 5.2.** *Let $\mathcal{DAG}$ denote the set of all directed acyclic graphs (DAGs). Let $\mathcal{H} := \{\mathcal{G}_\theta \mid \theta \in \Theta\}$ be the set of graphs over concepts representable by H-CMR. Then:*

$$\mathcal{H} = \mathcal{DAG}$$

*That is,*

$$\forall\, G \in \mathcal{DAG}, \exists\, \theta \in \Theta : \mathcal{G}_\theta = G, \quad \forall \theta \in \Theta : \mathcal{G}_\theta \text{ is a DAG}$$

For the proofs, we refer to Appendix E.

### 5.2 INTERPRETABILITY

In sharp contrast to other CBMs, H-CMR offers interpretability not only for task prediction but also for concept prediction. Most CBMs model the concepts as conditionally independent given the input, leading to their direct prediction through an uninterpretable black-box mechanism. In H-CMR,

---

[5]We use straight-through estimation for the thresholding operator in Equation 4 and the sampling of $R$.

interpretability is achieved by representing concepts as the logical evaluation of a neurally selected rule. H-CMR provides two distinct forms of interpretability: *local* and *global*.

H-CMR provides local interpretability by making the logic rules used for predicting both concepts and tasks explicitly transparent to the human for a given input. Once these rules are selected for a given input instance, the remaining computation is inherently interpretable, as it consists of logical inference over the structure of the graph using these rules.

H-CMR enables a form of global interpretability, as all possible rules applicable for obtaining each concept and task prediction are stored transparently in the memory. First, this allows for human inspection of the rules, and even formal verification against predefined constraints using automated tools (see Appendix A). Second, this allows for *model interventions* (see Section 5.3).

## 5.3 INTERVENABILITY

**Concept interventions.** These are test-time operations where some concept predictions are replaced with ground truth values, simulating expert interaction at decision time. They are considered a key feature of CBMs, and ideally should maximally influence the model's predictions. Unlike CBMs that assume conditional independence, H-CMR allows interventions on parent concepts to propagate to child concepts, which in turn can influence further downstream concepts. As a result, H-CMR demonstrates greater responsiveness to interventions compared to CBMs modelling concepts as conditionally independent. Specifically, an intervention on a concept can affect child concept predictions in two ways. First, it can modify the rule selection process for the child concept, because the intervened concept is an input to the child concept's rule selection. Second, it can alter the evaluation of the selected logic rule, as the concept may be used in evaluating that rule.

**Model interventions.** In addition to concept interventions at test time, H-CMR allows for model interventions at training time, influencing the graph and rules that are learned. A human expert's knowledge can be incorporated by manually adding new rules to the memory, and learned rules can be inspected, modified, or replaced as needed. Moreover, the human can forbid concepts from being parents of other concepts, or enforce specific structures on the graph (e.g. choose which concepts should be sources or sinks). For details on how this can be done, we refer to Appendix A.

## 6 EXPERIMENTS

In our experiments, we consider the following research questions: **(Accuracy)** Does H-CMR attain similar concept accuracy as existing CBMs? Does H-CMR achieve high task accuracy irrespective of the concept set? **(Explainability and intervenability)** Does H-CMR learn meaningful rules? Are concept interventions effective? Can model interventions be used to improve data efficiency?

### 6.1 EXPERIMENTAL SETTING

We only list essential information for understanding the experiments. Details can be found in Appendix B. We focus on concept prediction as this is what distinguishes H-CMR the most, omitting the task for most experiments and comparing with state-of-the-art concept predictors. In one experiment we still support our claim that H-CMR can achieve high task accuracy irrespective of the concept set.

**Data and tasks.** We use four datasets to evaluate our approach: CUB (Welinder et al., 2010), a dataset for bird classification; MNIST-Addition (Manhaeve et al., 2018), a dataset based on MNIST (Lecun et al., 1998); CIFAR10 (Krizhevsky et al., 2009), a widely used dataset in machine learning; and a synthetic dataset based on MNIST with difficult concept prediction and strong inter-concept dependencies. All datasets except CIFAR10 provide full concept annotations. For CIFAR10, we use the same technique as Oikarinen et al. (2023b) to extract concept annotations from a vision-language model, showing our approach also works on non-concept-based datasets.[6]

**Evaluation.** We measure classification performance using accuracy. For intervenability, we report the effect on accuracy of intervening on concepts (see Appendix D for the used intervention strategies and ablation studies). Metrics are reported using the mean and standard deviation over 3 seeded runs.

---

[6]Note that our approach can be applied to multilabel classification dataset *without any concepts*, in which case H-CMR learns a graph over the different tasks (instead of over concepts and tasks).

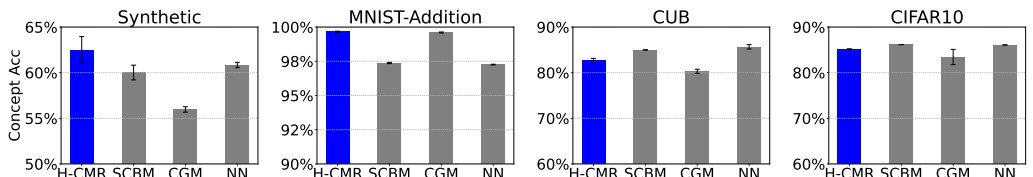

Figure 4: Concept accuracy for all datasets and models.

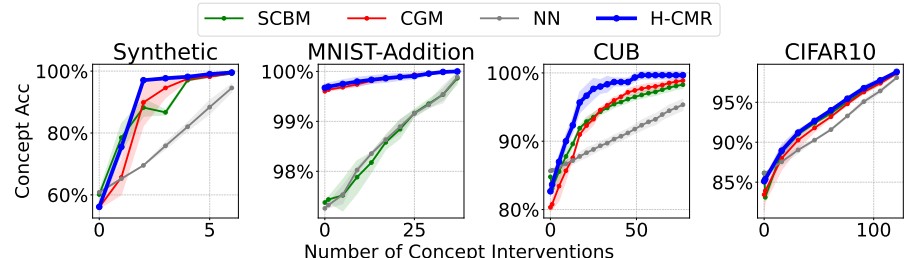

Figure 5: Concept accuracy before vs. after intervening on increasingly more concepts.

**Competitors.** We compare H-CMR with Stochastic Concept Bottleneck Models (SCBM) (Vanden-hirtz et al., 2024) and Causal Concept Graph Models (CGM) (Dominici et al., 2024), state-of-the-art CBMs developed with a strong focus on intervenability. We also compare with a neural network (NN) directly predicting the concepts, which is the concept predictor for most CBMs, such as Concept Bottleneck Models (CBNM) (Koh et al., 2020) and Concept-based Memory Reasoner (Debot et al., 2024). In the task accuracy experiment, we compare with SCBM, CGM, and CBNM.

## 6.2 KEY FINDINGS

**H-CMR's interpretability does not harm concept accuracy (Figure 4), and achieves high task accuracy irrespective of the concept set (Figure 7).** H-CMR achieves similar levels of concept accuracy compared to competitors. As a universal classifier, H-CMR can achieve high task accuracy even with small concept sets, similar to some other CBMs (Espinosa Zarlenga et al., 2022).

**H-CMR shows a high degree of intervenability (Figure 5).** We evaluate H-CMR's gain in concept accuracy after intervening on increasingly more concepts. H-CMR demonstrates far higher degrees of intervenability compared to CBMs modelling concepts independently (NN), and similar or better to approaches that model concepts dependently (SCBM, CGM).

**Model interventions by human experts improve data efficiency (Figure 6).** We exploit these to provide H-CMR with background knowledge about a subset of the concepts for MNIST-Addition, allowing H-CMR to maintain high accuracy even in low-data regimes with only partial supervision

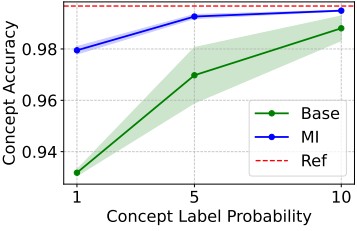

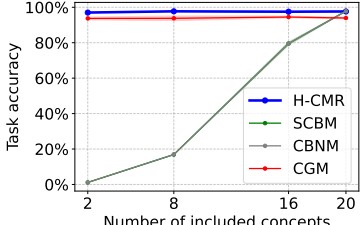

Figure 6: Data efficiency of H-CMR with (MI) and without (Base) background knowledge on MNIST-Add. The x-axis denotes how many concept labels are included in the training set. The reference is accuracy when training on all labels.

Figure 7: Task accuracy on MNIST-Addition for training on different sizes of the concept set. Universal classifiers (H-CMR, CGM) are robust to the choice of concepts, while other approaches are not (SCBM, CBNM).

on the concepts. We give H-CMR rules defining 18 concepts in terms of the 10 remaining ones, and force the latter to be source concepts. This helps in two ways. First, when source concepts are correctly predicted, so are the others. Second, when for a training example a label is only available on non-sources, the gradient can backpropagate through the given rules to provide a training signal to the sources. This means H-CMR does not only work in a concept-based setting, where full concept supervision is typically provided, but also in a neurosymbolic setting, where often *distant supervision* is used to train concepts by exploiting background knowledge (Manhaeve et al., 2018).

**H-CMR learns meaningful rules.** We qualitatively inspect the rules H-CMR learns in Appendix D.

## 7 RELATED WORK

H-CMR is related to two major directions in concept-based models (CBMs) research: one focusing on closing the *accuracy gap* between CBMs and black-box models like deep neural networks (Espinosa Zarlenga et al., 2022; Barbiero et al., 2023), and one focusing on *intervenability* (Espinosa Zarlenga et al., 2023; Havasi et al., 2022). The former has led to the development of many CBMs that are *universal classifiers*: they can achieve task accuracies comparable to black boxes irrespective of the concept set. However, many models achieve this by sacrificing the interpretability of their task predictions (Espinosa Zarlenga et al., 2022; Mahinpei et al., 2021). A notable exception is Concept Memory Reasoner (CMR), an interpretable universal classifier (Debot et al., 2024), achieved by modelling the task as the symbolic execution of a neurally selected logic rule from a memory.

However, the aforementioned CBMs treat concepts as conditionally independent, which limits the effect of *concept interventions* at test time: correcting a mispredicted concept only impacts the downstream task directly, not other (potentially correlated) concepts. To overcome this, a second line of work models dependencies between concepts (Havasi et al., 2022). For instance, Stochastic Concept Bottleneck Models (SCBMs) jointly model concepts rather than treating them independently (Vandenhirtz et al., 2024), enabling interventions to propagate across concepts. Yet, SCBMs are not universal classifiers: their accuracy is limited by the concept set. Causal Concept Graph Models (CGMs) address this by learning a concept graph and applying black-box message passing (Dominici et al., 2024), achieving both universality and high intervenability. However, the message passing makes concepts and tasks uninterpretable.

Table 1: CBMs having properties ($\checkmark$), partially ($\sim$) or not at all ($\times$): Universal Classifier (UC), Interpretable Predictions (IP), Expressive concept Interventions (EI), Model Interventions (MI).

| Model | UC | IP | EI | MI |
|---|---|---|---|---|
| CBNM | $\times$ | $\sim$ | $\times$ | $\sim$ |
| CEM | $\checkmark$ | $\times$ | $\times$ | $\times$ |
| CMR | $\checkmark$ | $\sim$ | $\times$ | $\sim$ |
| SCBM | $\times$ | $\sim$ | $\sim$ | $\times$ |
| CGM | $\checkmark$ | $\times$ | $\checkmark$ | $\sim$ |
| H-CMR | $\checkmark$ | $\checkmark$ | $\checkmark$ | $\checkmark$ |

Our model, H-CMR, can be seen as an extension of CMR's symbolic reasoning approach to concept predictions, combining it with CGMs' idea of concept graph learning. H-CMR achieves expressive interventions, and unlike previous models, it is a universal classifier that provides interpretability at both the concept and task levels. Table 1 gives an overview (for references, see Table 2).

We are also related to neurosymbolic approaches that perform rule learning. Some approaches operate on structured relational data such as knowledge graphs, where target predicates are predefined and a perception component is typically absent (Cheng et al., 2022; Qu et al., 2020). Others resemble standard CBMs, where the model's structure, i.e. which symbols ("tasks") are predicted from which others ("concepts"), is manually defined by the user (Si et al., 2019; Daniele et al., 2022; Tang & Ellis, 2023). Another line of work builds rules on input features instead of high-level concepts (Okajima & Sadamasa, 2019; Lee et al., 2022; 2025), and where rules typically provide local explanations (Lee et al., 2022; 2025). In contrast, H-CMR learns both the symbolic rules, which form both local and global explanations, and the dependency structure: a directed graph that defines how concepts and tasks depend on each other. Moreover, such works typically do not provide formal guarantees on expressivity, whereas we prove that H-CMR is a universal classifier.

## 8 CONCLUSION

We introduce H-CMR, a concept-based model that is a universal binary classifier while providing interpretability for both concept and task prediction. Through our experiments, we show that H-

CMR (1) achieves state-of-the-art concept and task accuracy, (2) is highly responsive to concept interventions at inference time, and (3) that through model interventions, background knowledge can be incorporated to improve data efficiency, if available. H-CMR can have societal impact by improving transparency and human-AI interaction.

**Limitations and future work.** Interesting directions for future work include extending the intervention strategy to take uncertainty into account, and performing a more extensive investigation of H-CMR's performance in a hybrid setting between concept-based and neurosymbolic, where for some concepts expert knowledge is available, and for others concept supervision. Furthermore, H-CMR's worst-case complexity is quadratic in the number of concepts, which is a limitation for very large concept sets.

**Reproducibility statement.** All our experiments are seeded, and we will make the code publicly available upon publication of the paper. Moreover, in Appendix B, we describe in detail the setup of each experiment, the implementation of each model, and the training setup.

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

# Supplementary Material

TABLE OF CONTENTS

Table 2: CBMs having properties (✓), partially (∼) or not at all (✗): Universal Classifier (UC), Interpretable Predictions (IP), Expressive concept Interventions (EI), Model Interventions (MI).

| Model | UC | IP | EI | MI |
|---|---|---|---|---|
| CBNM (Koh et al., 2020) | ✗ | ∼ | ✗ | ∼ |
| CEM (Espinosa Zarlenga et al., 2022) | ✓ | ✗ | ✗ | ✗ |
| CMR (Debot et al., 2024) | ✓ | ∼ | ✗ | ∼ |
| SCBM (Vandenhirtz et al., 2024) | ✗ | ∼ | ∼ | ✗ |
| CGM (Dominici et al., 2024) | ✓ | ✗ | ✓ | ∼ |
| H-CMR | ✓ | ✓ | ✓ | ✓ |

# A    DETAILS OF PROPERTIES IN H-CMR

## A.1    TASK PREDICTION

In H-CMR, tasks and concepts are modelled in the same way: they are nodes that are predicted from their parent nodes, and the learned rules in the memory define this structure. This means that the memory contains rules for predicting each concept, and for predicting each task. Consequently, the parametrization explained in the main text, which defines how concepts are predicted from other concepts, is also used for tasks. The simplest way to incorporate tasks is by simply considering them as additional concepts. Then, H-CMR learns a graph over concepts and tasks. This allows for instance that tasks are predicted using each other, and that concepts are predicted using tasks.

This approach is also possible in e.g. CGM (Dominici et al., 2024), but not in most CBMs, where concepts and tasks are modelled in two separate layers of the model. In most CBMs, concepts are first predicted from the input using a neural network, and then the task is predicted from the concepts (e.g. CBNM (Koh et al., 2020)) and possibly some residual, e.g. an embedding to provide additional contextual information (Mahinpei et al., 2021).

In H-CMR, this concept-task structure can be enforced by forcing the tasks to be sink nodes in the learned graph, i.e. they have no outgoing edges (no concepts are predicted from the tasks, and tasks are not predicted using each other), that additionally have all concepts as potential parents. This can be done through model interventions.

## A.2    MODEL INTERVENTIONS

In this section, we give some examples on how human experts can do model interventions on H-CMR during training, influencing the model that is being learned. To this end, we first derive the following matrix $A \in \mathbb{R}^{n_C \times n_C}$ from H-CMR's parametrization:

$$\forall i, j \in [1, n_C] : A_{ij} = \mathbb{1}[O_j > O_i] \tag{12}$$

where $A_{ij} = 1$ indicates that concept $j$ is allowed to be a parent of concept $i$ based on the node priority vector $O$ (see Equation 4). This matrix serves as an alternative representation of the parent-child constraints originally encoded by $O$, which can then be used in Equation 4 instead of $O$. By intervening on this matrix, it is possible to:

- force a concept $k$ to be a source (no parents): set $\forall k \in [1, n_C] : A_{kj} := 0$;
- force a concept $k$ to be a sink (no children): set $\forall k \in [1, n_C] : A_{jk} := 0$;
- forbid a concept $m$ to be a parent of concept $k$: set $A_{km} := 0$.

Furthermore, a specific topological ordering of the concepts can be enforced by explicitly assigning values to the entire node priority vector $O$. Moreover, to ensure that a concept $l$ precedes or follows concept $k$ in the topological ordering, one can set $O_l := O_k - z$ or $O_l := O_k + z$, respectively, where $z \in \mathbb{R}_0^+$ is any chosen positive number.

The human expert can also intervene on the roles the concepts play in individual rules. This means intervening on the 'unadjusted roles' $R'$, which are combined with the node priorities $O$ to form the rules. For instance, by intervening on $R'$ (and possibly $O$), it is possible to:

- force a concept $k$ to be absent in rule $l$ of concept $i$: set $p(R'_{i,l,k} = I) := 1$, $p(R'_{i,l,k} = P) := 0$, and $p(R'_{i,l,k} = N) := 0$;

- force a concept $k$ to be positively present in rule $l$ of concept $i$ (assuming $O$ allows it): set $p(R'_{i,l,k} = I) := 0$, $p(R'_{i,l,k} = P) := 1$, and $p(R'_{i,l,k} = N) := 0$;
- force a concept $k$ to be negatively present in rule $l$ of concept $i$ (assuming $O$ allows it): set $p(R'_{i,l,k} = I) := 0$, $p(R'_{i,l,k} = P) := 0$, and $p(R'_{i,l,k} = N) := 1$.

By intervening on the roles and the node priorities in these ways, experts can have fine-grained control over the content and structure of the rules. In the extreme case, human experts can choose to fully specify a rule, or even the entire rule set, through such interventions.

### A.3 VERIFICATION

Since the (learned) memory of rules is transparent, it can be formally verified against desired constraints in a similar fashion as for CMR (Debot et al., 2024). For instance, one can verify whether a constraint such as "*whenever the concept 'black wings' is predicted as True, the concept 'white wings' is predicted as False and the task 'pigeon' is predicted as False*" is guaranteed by the learned rules. This is possible because both concepts and tasks predictions can be represented as disjunctions over the rules in the memory, expressed in propositional logic. As described by debot2024interpretable, the neural rule selection can be encoded within this disjunction by introducing additional propositional atoms that denote whether each individual rule is selected, along with mutual exclusivity constraints between these atoms. Consequently, standard formal verification tools (e.g. model checkers) can be employed to verify constraints w.r.t. this propositional logic formula. We refer to Section 4.3 of debot2024interpretable.

## B EXPERIMENTAL AND IMPLEMENTATION DETAILS

**Datasets.** In CUB (Welinder et al., 2010), there are 112 concepts related to bird characteristics, such as wing pattern and head size. Each input consists of a single image containing a bird. In MNIST-Addition (Manhaeve et al., 2018), the input consists of two MNIST images (LeCun et al., 1998). There are 10 concepts per image, representing the digit present, and 19 tasks corresponding to the possible sums of the two digits. For CIFAR10, which does not include predefined concepts, we use the same technique as oikarinen2023labelfree to obtain them. Specifically, we use the same concept set as oikarinen2023labelfree, which they obtained by prompting an LLM, and obtain concept annotations by exploiting vision-language models, as in their work. For our synthetic dataset, we modify MNIST-Addition by restricting it to examples containing only the digits zero and one. We discard the original concepts and tasks and instead generate new concepts and their corresponding labels for each example using the following sampling process:

- $p(C_0 = 1) = \begin{cases} 0.7 & \text{if the first digit is a 1} \\ 0 & \text{otherwise} \end{cases}$

- $p(C_1 = 1) = \begin{cases} 0.7 & \text{if the second digit is a 1} \\ 0 & \text{otherwise} \end{cases}$

- $p(C_2 = 1 \mid \hat{c}_0, \hat{c}_1) = \hat{c}_0 \oplus' \hat{c}_1$
- $p(C_3 = 1 \mid \hat{c}_0, \hat{c}_2) = \hat{c}_0 \oplus' \hat{c}_2$
- $p(C_4 = 1 \mid \hat{c}_1, \hat{c}_2) = \hat{c}_1 \oplus' \hat{c}_2$
- $p(C_5 = 1 \mid \hat{c}_3, \hat{c}_4) = \hat{c}_3 \oplus' \hat{c}_4$
- $p(C_6 = 1 \mid \hat{c}_0, \hat{c}_1) = \hat{c}_0 \oplus' \hat{c}_1$

where $\oplus$ is the logical XOR, and where we define $\oplus'$ as a noisy XOR:

$$\hat{c}_i \oplus' \hat{c}_j = \begin{cases} 1 & \text{if } \hat{c}_i \oplus \hat{c}_j = 1 \\ 0.05 & \text{otherwise} \end{cases} \tag{13}$$

For each example, we sample labels from the above distributions. Intuitively, the concepts $C_0$ and $C_1$ indicate whether the corresponding MNIST images contain the digit one; however, these labels are intentionally noisy. The remaining concepts are constructed as noisy logical XORs of $C_0$, $C_1$, and of each other, introducing additional complexity and interdependence among the concepts.

**Reproducibility.** For reproducibility, we used seeds 0, 1 and 2 in all experiments.

**Model input.** For MNIST-Addition, CIFAR10 and the synthetic dataset, we train directly on the images. For CIFAR10, we use the same setup as vandenhirtz2024stochastic. For CUB, we instead use pretrained Resnet18 embeddings (He et al., 2016), using the setup of debot2024interpretable.

**General training information.** We use the AdamW optimizer. For H-CMR and CBNMs, we maximize the likelihood of the data. SCBMs and CGMs are trained using their custom loss functions. After training, we select the model checkpoint with the highest validation accuracy. Validation accuracy refers to concept prediction accuracy in all experiments, except in the MNIST-Addition setting where tasks are retained. In that case, accuracy is computed over the concatenation of both concepts and tasks. Throughout, we model all concepts and tasks as independent Bernoulli random variables.

**Intervention policy.** For the results presented in the main text, the intervention policy follows the graph learned by H-CMR, intervening first on the sink nodes and gradually moving down the topological ordering as determined by the learned node priority vector. This approach makes it easy to interpret the results, as the intervention order is the same for different models (i.e. if we intervene on 3 concepts, they are the same concepts for H-CMR and all competitors). This makes it clear how well the models are able to *propagate* the intervention to other concept predictions. To ensure a fair comparison with CGM, we make sure that CGM learns using the same graph that H-CMR learned. Additional ablation studies using different intervention policies are provided in Appendix D.

**General architectural details.** For each experiment, we define a neural network $\phi$ that maps the input to some latent embedding with size $size_{latent}$ a hyperparameter. The architecture of this neural network depends on the experiment but is the same for all models.

In H-CMR's encoder, $f_i$ (see Equation 1) is implemented as first applying $\phi$ to the input, producing a latent embedding with size $size_{latent}$. This passes through a linear layer with leaky ReLU activation outputting 2 embeddings of size $size_{c\,emb}$ per concept (similar to concept embeddings (Espinosa Zarlenga et al., 2022), we call one embedding the "positive" one, and the other one the "negative" one), with $size_{c\,emb}$ a hyperparameter. For each concept, the two embeddings are concatenated and the result passes through a linear layer with sigmoid activation to produce the concept prediction probability (corresponding to $f_i$ in Equation 1. The concatenation of all embeddings form the output embedding of the encoder ($\hat{e}$ as produced by $g$ in Equation 1).

For H-CMR's decoder, the neural selection $p(S_i = k \mid \hat{e}, \hat{c}_{parents(i)})$ is implemented in the following way. First, we mask within $\hat{e}$ the values that originally corresponded to non-parent concepts. Then, for each parent concept, if it is predicted to be True, we mask the values of $\hat{e}$ that correspond to its negative embedding. If it is predicted to be False, we mask the ones corresponding to its positive embedding. Note again that this is similar to concept embeddings (Espinosa Zarlenga et al., 2022). Next, we concatenate the embedding with the concept predictions $\hat{c}$. For non-parents, we set their value always to 0. The result is a tensor of shape $2 \cdot n_c \cdot size_{c\,emb} + n_C$, which is passed through a linear layer (ReLU activation) with output size $size_{latent}$. This is then passed through another linear layer with output size $n_C \cdot n_R$, which is reshaped to the shape $(n_C, n_R)$. For each row $i$ in this tensor, this represents the logits of $p(S_i \mid \cdot)$. We apply a softmax over the last dimension to get the corresponding probabilities.

In H-CMR's memory, the node priority vector $O$ is implemented as a torch Embedding of shape $(1, n_C)$. The rule embeddings in the memory (each representing the latent representation of a logic rule) are implemented as a torch Embedding of shape $(n_R \cdot n_C, size_{rule\,emb})$ with $size_{rule\,emb}$ a hyperparameter. This is reshaped to shape $(n_C, n_R, size_{rule\,emb})$. The "rule decoding" neural network consists of a linear layer (leaky ReLU) with output size $size_{rule\,emb}$ followed by a linear layer with output size $n_C \cdot n_R$. After passing the embedding through this neural network, the result has output shape $(n_C, n_R, n_C \cdot 3)$, which is reshaped into $(n_C, n_R, n_C, 3)$. This corresponds to the logits of the 'unadjusted roles' $p(R')$. We obtain the probabilities by applying a softmax to the last dimension.

The CBNM first applies $\phi$, and then continues with a linear layer with sigmoid activation outputting the concept probabilities. The task predictor is a feed-forward neural network consisting of 2 hidden layers with ReLU activation and dimension 100, and a final linear layer with sigmoid activation. We employ hard concepts to avoid the problem of concept leakage which may harm interpretability

(Marconato et al., 2022), meaning we threshold the concept predictions at 50% before passing them to the task predictor.

For SCBM, we use the implementation as given by the authors.[7] The only differences are that (1) we use $\phi$ to first produce a latent embedding which we pass to SCBM, and (2) we replace the softmax on their task prediction by a sigmoid, as we treat the tasks in all experiments independently. For the $\alpha$ hyperparameter, we always use 0.99. We always use the amortized variant of SCBMs, which is encouraged by vandenhirtz2024stochastic. We use 100 Monte Carlo samples.

For CGM, we also use the implementation as given by the authors.[8] The only difference is that we use $\phi$ to first produce a latent embedding, which is passed to CGM. To ensure a fair comparison when measuring intervenability, we equip CGM with the same graph that H-CMR learned.

**Hyperparameters per experiment.** In CUB, $\phi$ is a feed-forward neural network with 3 hidden layers. Each layer has output size $size_{latent}$. We use a learning rate of 0.001, batch size 1048, and train for 500 epochs. We check validation accuracy every 20 epochs. For H-CMR, we use $size_{latent} = 256$, $size_{rule\,emb} = 500$, $size_{c\,emb} = 10$, $n_R = 5$, and $\beta = 0.1$. For SCBM, we use $size_{latent} = 64$. For CGM, we use $size_{latent} = 64$ and $size_{c\,emb} = 8$.

In the synthetic dataset, $\phi$ consists of a convolutional neural network (CNN) consisting of a Conv2d layer (1 input channel, 6 output channels and kernel size 5), a MaxPool2d layer (kernel size and stride 2), a ReLU activation, a Conv2D layer (16 output channels and kernel size 5), another MaxPool2d layer (kernel size and stride 2), another ReLU activation, a flattening layer and finally a linear layer with output size $size_{latent}/2$. $\phi$ applies this CNN to both input images and concatenates the resulting embeddings. We use a learning rate of 0.001, batch size 256, and train for 100 epochs. We check validation accuracy every 5 epochs. For H-CMR, we use $size_{latent} = 128$, $size_{rule\,emb} = 1000$, $size_{c\,emb} = 3$, $n_R = 10$, and $\beta = 0.1$. For SCBM, we use $size_{latent} = 128$. For CGM, we use $size_{latent} = 128$ and $size_{c\,emb} = 3$. For CBNM, we use $size_{latent} = 128$.

In MNIST-Addition, $\phi$ consists of a CNN that is the same as for the synthetic dataset, but with 3 additional linear layers at the end with output size $size_{latent}/2$, the first two having ReLU activation. $\phi$ applies this CNN to both input images and concatenates the resulting embeddings. We use a learning rate of 0.001, batch size 256, and train for 300 epochs. We check validation accuracy every 5 epochs. For H-CMR, we use $size_{latent} = 128$, $size_{rule\,emb} = 1000$, $size_{c\,emb} = 3$, $n_R = 10$, and $\beta = 0.1$. For SCBM, we use $size_{latent} = 256$. For CGM, we use $size_{latent} = 100$ and $size_{c\,emb} = 8$. For CBNM, we use $size_{latent} = 128$.

In CIFAR10, $\phi$ consists of a CNN consisting of a Conv2d layer (3 input channels, 32 output channels, kernel size 5 and stride 3), ReLU activation, a Conv2d layer (32 input channels, 64 output channels, kernel size 5 and stride 3), ReLU activation, a MaxPool2d layer (kernel size 2), a Dropout layer (probability of 50%), a flattening layer and a linear layer with output size $size_{latent}$ with ReLu activation. We use a learning rate of 0.001, batch size 100, and train for 300 epochs. We check validation accuracy every 5 epochs. For H-CMR, we use $size_{latent} = 100$, $size_{rule\,emb} = 500$, $size_{c\,emb} = 3$, $n_R = 10$, and $\beta = 0.1$. For SCBM, we use $size_{latent} = 100$. For CGM, we use $size_{latent} = 100$ and $size_{c\,emb} = 8$. For CBNM, we use $size_{latent} = 100$.

**Additional details of the MNIST-Addition experiments.** In the MNIST-Addition experiment where only concept accuracy is reported, we treat the labels for both the individual digits and their sums as concepts. For the experiment reporting task accuracy, we use the more conventional setting, where the digits are considered as concepts, while the sums of digit pairs are treated as tasks. There, we use model interventions to make the tasks sinks in the graph.

**Hyperparameter search.** The hyperparameters for all models were chosen that result in the highest validation accuracy. For $size_{latent}$, we searched within the grid $[32, 64, 100, 128, 200, 256, 512]$, for $size_{c\,emb}$ within $[2, 3, 8, 16]$, and for $size_{rule\,emb}$ within $[100, 500, 1000]$.

**Prototypicality regularization.** Similar to the approach of CMR (Debot et al., 2024), we employ a regularization term that encourages the learned rules to be more *prototypical* of the seen concepts during training. This aligns with standard theories in cognitive science (Rosch, 1978). While this notion of prototypicality has inspired many so-called prototype-based models (Rudin, 2019; Li et al.,

---

[7]https://github.com/mvandenhi/SCBM
[8]https://github.com/gabriele-dominici/CausalCGM

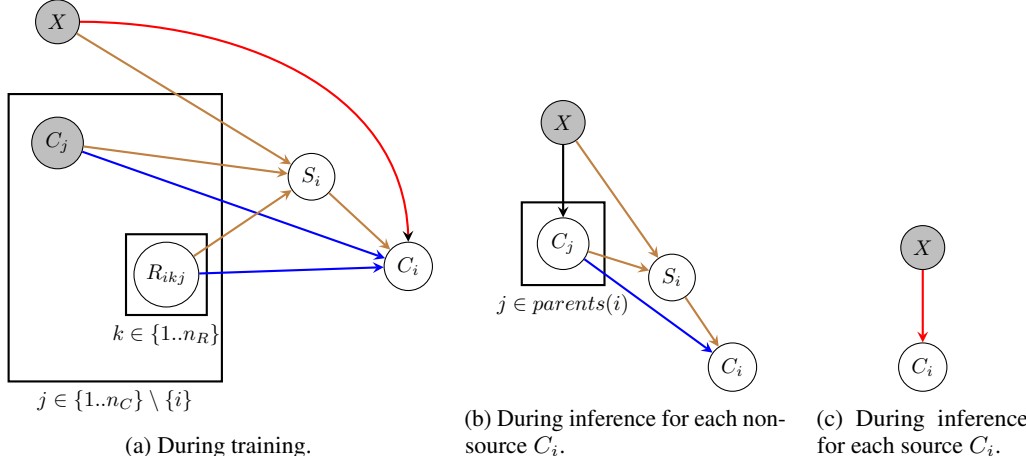

(a) During training.

(b) During inference for each non-source $C_i$.

(c) During inference for each source $C_i$.

Figure 8: Part of the probabilistic graphical model for computing a single node $C_i$. Red edges denote the prediction by the encoder for source concepts; brown edges denote the rule selection; blue edges denote the rule evaluation. Grey nodes are observed. During inference, the roles are always observed and fixed, so we do not write them. The observed roles determine which concepts are parents of $C_i$, and whether $C_i$ is a source. For each parent $C_j$, the black edge is a "nested" (b) or (c), depending on whether $C_j$ is a source.

2018; Chen et al., 2019), where prototypes are built in the input space, such as images. Just like CMR, H-CMR differs from such models significantly. For instance, H-CMR gives a logical interpretation to prototypes as being logical rules. Moreover, the prototypes are built in the concept space (as opposed to the input space). We refer to debot2024interpretable (specifically, Section 4.2) for more such differences. During training, this regularization is present as an additional factor in the decoder, replacing Equation 2 with

$$p(C_i = 1 \mid \hat{e}, \hat{c}, \hat{r}_i, \hat{y}) = \sum_{k=1}^{n_R} \underbrace{p(S_i = k \mid \hat{e}, \hat{c}_{parents(i)})}_{\substack{\text{neural selection of rule } k \\ \text{using parent concepts + emb.}}} \cdot \underbrace{l(\hat{c}_{parents(i)}, \hat{r}_{i,k})}_{\substack{\text{evaluation of concept } i\text{'s rule } k \\ \text{using parent concepts}}} \cdot \underbrace{p_{reg}(r_{i,k} = \hat{c})^{\beta \cdot \hat{y}}}_{\substack{\text{prototypicality of} \\ \text{concept } i\text{'s rule } k}}$$

(14)

where $\beta$ is a hyperparameter, and

$$p_{reg}(r_{i,k} = \hat{c}) = \prod_{j=1}^{n_C} (0.5 \cdot \mathbb{1}[r_{i,k,j} = I] + \mathbb{1}[\hat{c}_j = 1] \cdot \mathbb{1}[r_{i,k,j} = P] + \mathbb{1}[\hat{c}_j = 0] \cdot \mathbb{1}[r_{i,k,j} = N])$$

(15)

Inuitively, the rule that is selected should not only provide a correct prediction, but also resemble the seen concepts as much as possible. For instance, if a concept is True, the loss prefers to see a positive role, over irrelevance, over a negative one. Like in (Debot et al., 2024), the regularization only affects positive training examples ($\hat{y} = 1$).

## C PROBABILISTIC GRAPHICAL MODEL

### C.1 GENERAL

In Figure 8, we give the probabilistic graphical model for H-CMR during training (Figure 8a), during inference for non-source concepts (Figure 8b) and for source concepts (Figure 8c). Additionally, we provide an 'extended PGM' in Figure 9, where we add the additional variables that we use in some of the equations (which are effectively marginalized out in Figure 8).

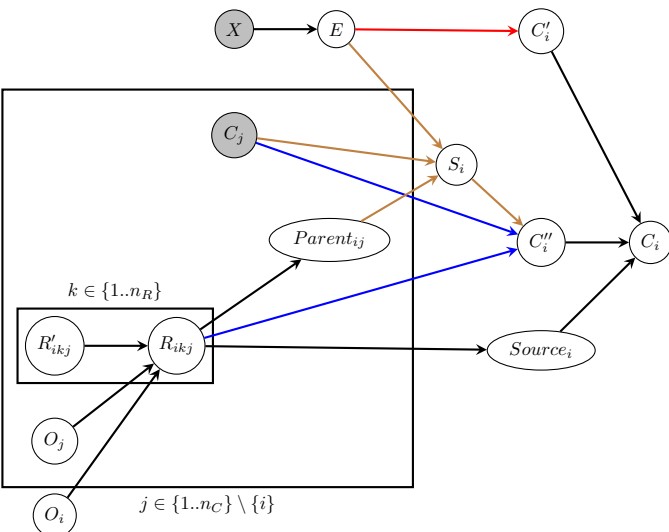

Figure 9: Part of the probabilistic graphical model for computing a single node $C_i$ during training, extended with additional variables. $E$ is an embedding represented by a delta distribution. $Parent_{ij}$ is a Bernoulli denoting whether $j$ is a parent of $i$. $C_i'$ denotes prediction of the concept by the encoder, while $C_i''$ is prediction using the rules in the memory. $Source_i$ is a Bernoulli denoting whether $i$ is a source concept, and serves as a "selection" between $C_i'$ and $C_i''$. Each $O_j$ is a delta distribution enforcing a topological ordering.

We choose the following conditional probabilities:

$$p(E \mid \hat{x}) = \delta(E - \hat{e}) \quad \text{where} \quad \hat{e} = g(\hat{x}) \tag{16}$$

$$p(\hat{c}_i \mid \hat{c}_i'', \hat{c}_i', so\hat{u}rce_i) = \mathbb{1}[so\hat{u}rce_i = 1] \cdot \mathbb{1}[\hat{c}_i' = \hat{c}_i] + \mathbb{1}[so\hat{u}rce_i = 0] \cdot \mathbb{1}[\hat{c}_i'' = \hat{c}_i] \tag{17}$$

$$p(C_i' = 1 \mid \hat{e}) = f_i(\hat{e}) \tag{18}$$

$$p(C_i'' = 1 \mid S_i = k, \hat{c}, \hat{r}_{i,k}) = l(\hat{c}_{parents(i)}, \hat{r}_{i,k}) \tag{19}$$

$$p(S_i = k \mid \hat{x}, \hat{c}, pa\hat{r}ent_i) = h_{i,k}(\hat{x}, \{\hat{c}_j \mid pa\hat{r}ent_{ij} = 1\}) \tag{20}$$

$$p(Source_i = 1 \mid \hat{r}_i) = \prod_{j=1}^{n_C} \prod_{k=1}^{n_R} \mathbb{1}[\hat{r}_{i,k,j} = I] \tag{21}$$

$$p(Parent_{ij} = 1 \mid \hat{r}_{i,:,j}) = 1 - \prod_{k=1}^{n_R} \mathbb{1}[\hat{r}_{i,k,j} = I] \tag{22}$$

$$p(R_{i,k,j} = P \mid \hat{r}_{i,k,j}', \hat{o}_i, \hat{o}_j) = \mathbb{1}[\hat{o}_j > \hat{o}_i] \cdot \mathbb{1}[\hat{r}_{i,k,j}' = P] \tag{23}$$

$$p(R_{i,k,j} = N \mid \hat{r}_{i,k,j}', \hat{o}_i, \hat{o}_j) = \mathbb{1}[\hat{o}_j > \hat{o}_i] \cdot \mathbb{1}[\hat{r}_{i,k,j}' = N] \tag{24}$$

$$p(R_{i,k,j} = I \mid \hat{r}_{i,k,j}', \hat{o}_i, \hat{o}_j) = \mathbb{1}[\hat{o}_j > \hat{o}_i] \cdot \mathbb{1}[\hat{r}_{i,k,j}' = I] + \mathbb{1}[\hat{o}_j \leq \hat{o}_i] \tag{25}$$

$$p(O_j) = \delta(O_j - \hat{o}_j) \quad \text{where} \quad \hat{o}_j \text{ is a learnable parameter} \tag{26}$$

where $f_i$, $h_{i,k}$ and $g$ are neural networks parametrizing the logits of a Bernoulli, the logits of a categorical, and the point mass of a delta distribution, respectively. The remaining probability is the categorical distribution $p(R_{i,k,j}')$, whose logits are parametrized by a learnable embedding and a neural network mapping this embedding on the logits.

## C.2 DERIVING EQUATION 10

Now we will derive the likelihood formula during training (Equation 10). For brevity, we will abbreviate $Source$ as $Src$ and we denote with $\hat{c}$ an assignment to all concepts *except* $C_i$. When summing over $\hat{r}$, we mean summing over all possible assignments to these variables, which are $n_C$

categoricals each with 3 possible values. First we marginalize out $Src$, $C_i'$ and $C_i''$:

$$p(\hat{c}_i \mid \hat{c}, \hat{x}) = \sum_{s\hat{r}c_i=0}^{1} \sum_{\hat{c}_i''=0}^{1} \sum_{\hat{c}_i'=0}^{1} p(\hat{c}_i \mid s\hat{r}c_i, \hat{c}_i', \hat{c}_i'') \cdot p(s\hat{r}c_i, \hat{c}_i', \hat{c}_i'' \mid \hat{c}, \hat{x}) \tag{27}$$

We now marginalize $E$ and $R$, exploiting the conditional independencies that follow from the PGM:

$$p(\hat{c}_i \mid \hat{c}, \hat{x}) = \int_{\hat{e}} \sum_{\hat{r}} \sum_{s\hat{r}c_i=0}^{1} \sum_{\hat{c}_i''=0}^{1} \sum_{\hat{c}_i'=0}^{1} p(\hat{c}_i \mid s\hat{r}c_i, \hat{c}_i', \hat{c}_i'') \cdot p(s\hat{r}c_i, \hat{c}_i', \hat{c}_i'' \mid \hat{c}, \hat{x}, \hat{e}, \hat{r}) \cdot p(\hat{e}|\hat{x}) \cdot p(\hat{r}) \, \mathrm{d}\hat{e}$$
$$\tag{28}$$

Given $C$, $X$, $E$ and $R$, it follows from the PGM that $Src_i$, $C_i'$ and $C_i''$ are conditionally independent, and each of the resulting conditional probabilities can be simplified:

$$p(\hat{c}_i \mid \hat{c}, \hat{x}) = \int_{\hat{e}} \sum_{\hat{r}} \sum_{s\hat{r}c_i=0}^{1} \sum_{\hat{c}_i''=0}^{1} \sum_{\hat{c}_i'=0}^{1} p(\hat{c}_i \mid s\hat{r}c_i, \hat{c}_i', \hat{c}_i'') \cdot p(s\hat{r}c_i \mid \hat{r}) \cdot p(\hat{c}_i' \mid \hat{e}) \cdot p(\hat{c}_i'' \mid \hat{c}, \hat{e}, \hat{r})$$
$$\cdot p(\hat{e}|\hat{x}) \cdot p(\hat{r}) \, \mathrm{d}\hat{e} \tag{29}$$

We exploit the fact that $p(E \mid \hat{x})$ is a delta distribution. After applying the delta distribution's sifting property, we obtain:

$$p(\hat{c}_i \mid \hat{c}, \hat{x}) = \sum_{\hat{r}} \sum_{s\hat{r}c_i=0}^{1} \sum_{\hat{c}_i''=0}^{1} \sum_{\hat{c}_i'=0}^{1} p(\hat{c}_i \mid s\hat{r}c_i, \hat{c}_i', \hat{c}_i'') \cdot p(s\hat{r}c_i \mid \hat{r}) \cdot p(\hat{c}_i' \mid \hat{e}) \cdot p(\hat{c}_i'' \mid \hat{c}, \hat{e}, \hat{r}) \cdot p(\hat{r})$$
$$\tag{30}$$

with $\hat{e} = g(\hat{x})$ the point mass of the distribution. After filling in the conditional probability for $p(C_i \mid s\hat{r}c_i, \hat{c}_i', \hat{c}_i'')$, we have:

$$p(\hat{c}_i \mid \hat{c}, \hat{x}) = \sum_{\hat{r}} \sum_{s\hat{r}c_i=0}^{1} \sum_{\hat{c}_i''=0}^{1} \sum_{\hat{c}_i'=0}^{1} (\mathbb{1}[s\hat{r}c_i = 1] \cdot \mathbb{1}[\hat{c}_i' = \hat{c}_i] + \mathbb{1}[s\hat{r}c_i = 0] \cdot \mathbb{1}[\hat{c}_i'' = \hat{c}_i]) \tag{31}$$

$$\cdot p(\hat{c}_i \mid s\hat{r}c_i, \hat{c}_i', \hat{c}_i'') \cdot p(s\hat{r}c_i \mid \hat{r}) \cdot p(\hat{c}_i' \mid \hat{e}) \cdot p(\hat{c}_i'' \mid \hat{c}, \hat{e}, \hat{r}) \cdot p(\hat{r})$$
$$\tag{32}$$

Most terms become zero due to the indicator functions. After simplifying, we have:

$$p(\hat{c}_i \mid \hat{c}, \hat{x}) = \sum_{\hat{r}} p(\hat{r}) \cdot [p(Src_i = 1 \mid \hat{r}) \cdot p(C_i' = \hat{c}_i \mid \hat{e}) + p(Src_i = 0 \mid \hat{r}) \cdot p(C_i'' = \hat{c}_i \mid \hat{c}, \hat{e}, \hat{r})]$$
$$\tag{33}$$

$$= \mathbb{E}_{\hat{r} \sim p(R)} [p(Src_i = 1 \mid \hat{r}) \cdot p(C_i' = \hat{c}_i \mid \hat{e}) + p(Src_i = 0 \mid \hat{r}) \cdot p(C_i'' = \hat{c}_i \mid \hat{c}, \hat{e}, \hat{r})]$$
$$\tag{34}$$

which corresponds to Equation 10.

## C.3 DERIVING EQUATION 6

Now we will derive the likelihood formula used during inference (Equation 6). We begin by marginalizing out $\hat{c}$:

$$p(C_i \mid \hat{x}) = \sum_{\hat{c}} p(C_i \mid \hat{c}, \hat{x}) \cdot p(\hat{c} \mid \hat{x}) \tag{35}$$

where the sum goes over all possible assignments to all concepts (except $C_i$). As explained in Section 3, at inference time, we collapse the distributions over roles $p(R)$ on their most likely values $\hat{r}$, which means that each role $R_{i,k,j}$ is observed and fixed. After filling in Equation 10:

$$p(C_i \mid \hat{x}) = \sum_{\hat{c}} (\mathbb{1}[s\hat{r}c_i = 1] \cdot p(C_i' \mid \hat{e}) + \mathbb{1}[s\hat{r}c_i = 0]) \cdot p(C_i'' \mid \hat{e}, \hat{c}, \hat{r})) \cdot p(\hat{c} \mid \hat{x}) \tag{36}$$

Table 3: Example learned rules for CUB, MNIST-Addition and CIFAR10.

| Example Rule | Intuition |
|---|---|
| black throat $\leftarrow$ ¬yellow throat | Mutually exclusive concepts |
| black throat $\leftarrow$ black upperparts $\wedge$ ¬yellow throat $\wedge$ brown forehead | E.g. crows, ravens, blackbirds |
| $\text{digit}_1$ is 0 $\leftarrow$ ¬$\text{digit}_1$ is 1 $\wedge$ ¬$\text{digit}_1$ is 2 $\wedge$ ... | Mutually exclusive concepts |
| sum is 7 $\leftarrow$ $\text{digit}_1$ is 5 $\wedge$ $\text{digit}_2$ is 2 $\wedge$ ¬$\text{digit}_1$ is 1 $\wedge$ ... | $5 + 2 = 7$ |
| a dock $\leftarrow$ a port $\wedge$ a tire | Ports have docks |
| a passenger $\leftarrow$ a cab for the driver | Cabs and their passengers |
| engines $\leftarrow$ four wheels | Cars |
| hooves $\leftarrow$ long neck | Horses |

where $s\hat{r}c_i$ can be deterministically computed from $\hat{r}$. First, we use that $p(C_i' \mid \hat{e})$ is equivalent to $p(C_i' \mid \hat{x})$ as $E$ is a deterministic function of $X$:

$$p(C_i \mid \hat{x}) = \sum_{\hat{c}} (\mathbb{1}[s\hat{r}c_i = 1] \cdot p(C_i' \mid \hat{x}) + \mathbb{1}[s\hat{r}c_i = 0]) \cdot p(C_i'' \mid \hat{e}, \hat{c}, \hat{r})) \cdot p(\hat{c} \mid \hat{x}) \quad (37)$$

Then, we remark that the first term is independent of $\hat{c}$, and the second term is only dependent on $\hat{c}_{parents(i)}$, as follows from the conditional probabilities $p(C_i'' \mid S_i = k, \hat{c}, \hat{r}_{i,k})$ and $p(S_i \mid \hat{x}, \hat{c}, \hat{parent}_i)$. Then, we can rewrite the above equation as:

$$p(C_i|\hat{x}) = \sum_{\hat{c}_{parents(i)}} (\mathbb{1}[s\hat{r}c_i = 1] \cdot p(C_i' \mid \hat{x})$$
$$+ \mathbb{1}[s\hat{r}c_i = 0] \cdot p(C_i'' \mid \hat{x}, \hat{c}_{parents(i)}, \hat{r}_i)) \cdot p(\hat{c}_{parents(i)} \mid \hat{x}) \quad (38)$$

which corresponds to Equation 6.

# D ADDITIONAL RESULTS

## D.1 LEARNED RULES

**H-CMR learns meaningful rules (Table 3).** For instance, the listed rules for CUB show that H-CMR has learned that certain concepts are mutually exclusive (e.g. 'black throat' and 'yellow throat' for birds). In MNIST-Addition, H-CMR has also learned that a single MNIST image only contains 1 digit, and the rules of addition are also recognizable.

## D.2 DIFFERENT INTERVENTION POLICIES

In this ablation study, we investigate H-CMR's intervenability when using different intervention policies. We consider three policies:

- **Graph-based policy:** this policy is based on H-CMR's learned concept graph, first intervening on the source concepts and gradually moving to the sinks. This is a natural choice as intervening on earlier concepts in the graph may have a larger impact. This approach is also used by dominici2024causal for CGMs, and makes it intuitively easy to interpret results, as the intervention order is the same for different models (i.e. if we intervene on 3 concepts, they are the same concepts for H-CMR and all competitors). To ensure the comparison with CGMs is fair, we make sure that the CGMs learn using the same graph that H-CMR learned.

- **Uncertainty-based policy:** this policy uses the uncertainty of the concept predictions, first intervening on concepts with high uncertainty. This policy is introduced by vandenhirtz2024stochastic for SCBMs. Note that the intervention order differs between different input examples, and differs between different models.

- **Random policy:** this policy randomly generates an intervention order.

We report concept accuracy after different numbers of interventions (like in the main text). For the results in the paper, we additionally report the difference in accuracy after versus before intervening,

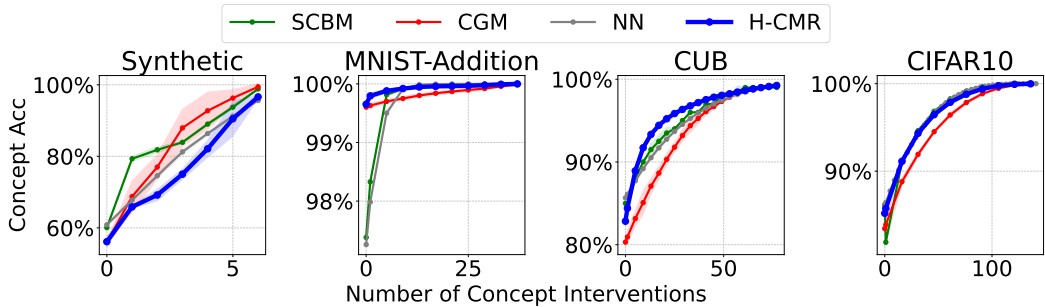

Figure 10: Concept accuracy using the uncertainty-based policy after intervening on increasingly more concepts.

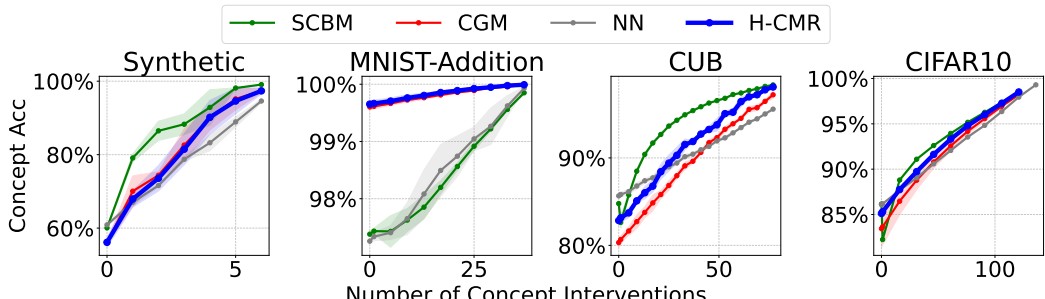

Figure 11: Concept accuracy using the random policy after intervening on increasingly more concepts.

measured on *non-intervened concepts*, after different numbers of interventions (Figure 12). The former is reported by e.g. vandenhirtz2024stochastic for SCBM, while the latter is reported by e.g. dominici2024causal for CGM.

Note that in the main text, we report concept accuracy using the graph-based policy.

**H-CMR performs well using other intervention policies (Figures 10 and 11).** Using SCBM's uncertainty-based policy (Figure 10), H-CMR performs better than competitors except for the synthetic dataset. When using a random policy (Figure 11), H-CMR performs similar to CGM. For instance, on CUB, H-CMR outperforms NN and CGM but is outperformed by SCBM. On other datasets except the synthetic one, H-CMR performs similar to SCBM and CGM. Note that, as SCBM does not predict concept in a hierarchical fashion, unlike CGM and H-CMR, SCBM is not inherently at a disadvantage when using a random policy.

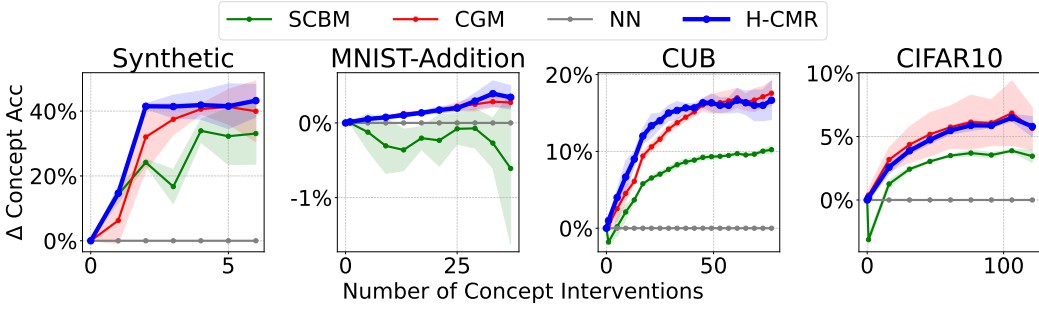

Figure 12: Difference in accuracy on non-intervened concepts using the graph-based policy after intervening on increasingly more concepts.

# E PROOFS

## E.1 THEOREM 5.1

We prove that H-CMR is a universal binary classifier if $n_R \geq 2$ in a similar fashion as done by debot2024interpretable for CMR.

*Proof.* Source concepts are directly predicted from the input $X$ using a neural network, which is a universal binary classifier. For non-source concepts and tasks $C_i$, the prediction is made by using a neural network to select a logic rule based on the parent concepts $C_{parents(i)}$ and the input $X$, which is then evaluated on the parent concepts $C_{parents(i)}$. Let $C_k$ be an arbitrarily chosen parent of $C_i$. Consider the following two rules, which can be expressed in H-CMR as shown between parentheses:

$$C_i \leftarrow C_k \quad \text{(i.e. } \hat{r}_{i,0,k} = P, \quad \forall j \neq k : \hat{r}_{i,0,j} = I) \tag{39}$$

$$C_i \leftarrow \neg C_k \quad \text{(i.e. } \hat{r}_{i,1,k} = N, \quad \forall j \neq k : \hat{r}_{i,1,j} = I) \tag{40}$$

By selecting one of these two rules, the rule selector neural network can always make a desired prediction for $C_i$ based on the predicted parent concepts $C_{parents(i)}$ and the input $X$. To predict $C_i = 1$, the first rule can be selected if $C_k$ is predicted *True*, and the second rule if it is predicted *False*. To predict $C_i = 0$, the opposite rule can be selected: the first rule if $C_k$ is predicted *False*, and the second rule if it is predicted *True*. $\square$

## E.2 THEOREM 5.2

For proving this theorem, we have to prove two statements: that H-CMR's parametrization can represent any DAG, and that H-CMR's parametrization guarantees that the directed graph implied by the rules is acyclic.

**Any DAG over the concepts can be represented by H-CMR:**

$$\forall G \in \mathcal{DAG}, \exists \theta \in \Theta : \mathcal{G}_\theta = G \tag{41}$$

*Proof.* We will show that for any graph $G$ over the concepts, there exists a set of parameters for H-CMR such that H-CMR has that graph. This means we need to prove that any possible set of edges that form a DAG, can be represented by H-CMR's parametrization.

Let $E_G$ be $G$'s set of edges. For any $G$, it is known that we can find a topological ordering $T_G$ of $G$'s nodes such that for every edge $(i,j) \in E_G$, $i$ occurs earlier in the ordering than $j$. Let $index(i, T_G)$ denote the index of $i$ in this ordering.

The following random variables of H-CMR are parameters that define the concept graph: $O \in \mathbb{R}^{n_C}$ (delta distributions, parametrized by an embedding), $R'_{i,k,j} \in \{P, N, I\}$ (categorical variables, parametrized by embeddings and neural networks).

The following assignment to these variables ensures that H-CMR's concept graph corresponds to $G$:

$$\forall i \in [1, n_C] : O_i = index(i, T_G) \tag{42}$$

$$\forall i, j \in [1, n_C], k \in [1, n_R] : p(R'_{i,k,j} = P) = \begin{cases} 1 & \text{if } (i,j) \in E \\ 0 & \text{if } (i,j) \notin E \end{cases} \tag{43}$$

$$\forall i, j \in [1, n_C], k \in [1, n_R] : p(R'_{i,k,j} = N) = 0 \tag{44}$$

$$\forall i, j \in [1, n_C], k \in [1, n_R] : p(R'_{i,k,j} = I) = 1 - p(R'_{i,k,j} = P) \tag{45}$$

As the parent relation defines the edges in the graph, we need to prove that:

$$\forall i, j \in [1, n_C] : Parent_{ij} = 1 \Leftrightarrow (i,j) \in E \tag{46}$$

From H-CMR's parametrization, we know that:

$$Parent_{ij} = 1 - \prod_{k=1}^{n_R} \mathbb{1}[R_{i,k,j} = I]$$

Thus, it follows that:

$$Parent_{ij} = 1 \Leftrightarrow \exists k \in [1, n_R] : R_{i,k,j} \neq I \tag{47}$$

$$\Leftrightarrow \exists k \in [1, n_R] : I \neq \underset{r \in \{P,N,I\}}{\arg \max} \; p(R_{i,k,j} = r) \tag{48}$$

where we used Equation 8. After plugging this into Equation 46, we still need to prove that:

$$\forall i, j \in [1, n_C] : (\exists k \in [1, n_R] : I \neq \underset{r \in \{P,N,I\}}{\arg \max} \; p(R_{i,k,j} = r)) \Leftrightarrow (i, j) \in E \tag{49}$$

We first prove the $\Leftarrow$ direction. If $(i, j) \in E$, then $i$ must appear before $j$ in $T_G$. Therefore:

$$index(i, T_G) < index(j, T_G) \tag{50}$$

If we take Equation 4 and fill in Equation 43, we know that for all $k$:

$$p(R_{i,k,j} = P) = \mathbb{1}[O_j > O_i] \cdot p(R'_{i,k,j} = P) = \mathbb{1}[O_j > O_i] \tag{51}$$

$$p(R_{i,k,j} = N) = \mathbb{1}[O_j > O_i] \cdot p(R'_{i,k,j} = N) = 0 \tag{52}$$

$$p(R_{i,k,j} = I) = \mathbb{1}[O_j > O_i] \cdot p(R'_{i,k,j} = I) + \mathbb{1}[O_j \leq O_i] = \mathbb{1}[O_j \leq O_i] \tag{53}$$

Then, filling in Equation 42 and using Equation 50, we get

$$p(R_{i,k,j} = P) = \mathbb{1}[index(j, T_G) > index(i, T_G)] = 1 \tag{54}$$

$$p(R_{i,k,j} = N) = 0 \tag{55}$$

$$p(R_{i,k,j} = I) = \mathbb{1}[index(j, T_G \leq index(i, T_G] = 0 \tag{56}$$

which proves the $\Leftarrow$ direction after applying this to Equation 49. We now prove the $\Rightarrow$ direction by proving that

$$\forall i, j \in [1, n_C] : (i, j) \notin E \Rightarrow (\forall k \in [1, n_R] : I = \underset{r \in \{P,N,I\}}{\arg \max} \; p(R_{i,k,j} = r)). \tag{57}$$

Using Equation 4 and filling in Equation 42, we know that for all $k$:

$$p(R_{i,k,j} = P) = \mathbb{1}[O_j > O_i] \cdot p(R'_{i,k,j} = P) = 0 \tag{58}$$

$$p(R_{i,k,j} = N) = \mathbb{1}[O_j > O_i] \cdot p(R'_{i,k,j} = N) = 0 \tag{59}$$

$$p(R_{i,k,j} = I) = \mathbb{1}[O_j > O_i] \cdot p(R'_{i,k,j} = I) + \mathbb{1}[O_j \leq O_i] = \mathbb{1}[O_j > O_i] + \mathbb{1}[O_j \leq O_i] = 1 \tag{60}$$

which proves the $\Rightarrow$ direction. $\square$

**In H-CMR, the directed graph implied by the rules is always acylic:**

$$\forall \theta \in \Theta : \mathcal{G}_\theta \text{ is a DAG} \tag{61}$$

*Proof.* A graph $G$ is a DAG if and only if there exists a topological ordering $T_G$ for that graph, such that for every edge $(i, j) \in E_G$, $i$ occurs earlier in $T_G$ than $j$, with $E_G$ the set of edges of $G$. Thus, we must simply prove that there exists such a topological ordering for each possible $\theta \in \Theta$. We will prove that the node priority vector $O$ defines such an ordering.

Specifically, let $T_O := \arg \text{sort}(O)$ be the concepts sorted based on their value in $O$. We will prove that $T_O$ forms the topological ordering for $\mathcal{G}$.

We know that

$$T_O \text{ is a topological ordering for } \mathcal{G} \Leftrightarrow \forall (i, j) \in E_\mathcal{G} : O_i < O_j \tag{62}$$

As the parent relation defines the edges in the graph, we need to prove that:

$$\forall i, j \in [1, n_C] : Parent_{ij} = 1 \Rightarrow O_i < O_j \tag{63}$$

for which we will use a proof by contradiction. Let us assume that the above statement is false. Then, we know that:

$$\exists i, j \in [1, n_C] : Parent_{ij} = 1 \land O_i \geq O_j \tag{64}$$

From H-CMR's parametrization, we know that:

$$Parent_{ij} = 1 - \prod_{k=1}^{n_R} \mathbb{1}[R_{i,j,k} = I]$$

Thus, we know that:

$$\exists i, j \in [1, n_C], k \in [1, n_R] : R_{i,j,k} \neq I \wedge O_i \geq O_j \tag{65}$$

Then, using Equation 8, this is equivalent to:

$$\exists i, j \in [1, n_C], k \in [1, n_R] : I \neq \underset{r \in \{P, N, I\}}{\arg \max} \; p(R_{i,k,j} = r)) \wedge O_i \geq O_j \tag{66}$$

Filling in $O_i \geq O_j$ in Equation 4 gives:

$$p(R_{i,k,j} = P) = \mathbb{1}[O_j > O_i] \cdot p(R'_{i,k,j} = P) = 0 \tag{67}$$

$$p(R_{i,k,j} = N) = \mathbb{1}[O_j > O_i] \cdot p(R'_{i,k,j} = N) = 0 \tag{68}$$

$$p(R_{i,k,j} = I) = \mathbb{1}[O_j > O_i] \cdot p(R'_{i,k,j} = I) + \mathbb{1}[O_j \leq O_i] = 1 \tag{69}$$

which is a contradiction, as $I$ is the most likely role. $\square$

## F  LLM USAGE DECLARATION

During writing, Large Language Models (LLMs) were used only to polish and improve the clarity of the text.

## G  CODE, LICENSES AND RESOURCES

Our code will be made publicly available upon acceptance under the Apache license, Version 2.0. We implemented H-CMR in Python 3.10.12 and additionally used the following libraries: PyTorch v2.5.1 (BSD license) (Paszke et al., 2019), PyTorch-Lightning v2.5.0 (Apache license 2.0), scikit-learn v1.5.2 (BSD license) (Pedregosa et al., 2011), PyC v0.0.11 (Apache license 2.0). We used CUDA v12.7 and plots were made using Matplotlib (BSD license).

We used the implementation of Stochastic Concept Bottleneck Models (Apache license 2.0)[9] and Causal Concept Graph Models (MIT license)[10].

The used datasets are available on the web with the following licenses: CUB (MIT license[11]), MNIST (CC BY-SA 3.0 DEED), CIFAR10 (MIT license)[12].

The experiments were run on a machine with an NVIDIA GeForce RTX 3080 Ti, Intel(R) Xeon(R) Gold 6230R CPU @ 2.10GHz and 256 GB RAM. Table 4 shows the estimated total computation time for a single run of each experiment.

Table 4: Estimated total computation time for a single run of each experiment.

| Experiment | Time (hours) |
|---|---|
| CUB | 12.2 |
| MNIST-Addition | 3.3 |
| MNIST-Addition (with tasks) | 1.6 |
| CIFAR10 | 33.1 |
| Synth | 0.1 |

---

[9] https://github.com/mvandenhi/SCBM

[10] https://github.com/gabriele-dominici/CausalCGM

[11] https://huggingface.co/datasets/cassiekang/cub200_dataset

[12] https://www.kaggle.com/datasets/ekaakurniawan/the-cifar10-dataset/data

