# OpenReview forum: "Interpretable Hierarchical Concept Reasoning through Graph Learning"
_ICLR.cc/2026/Conference — Submitted to ICLR 2026_

### Official Review · Reviewer_3RNr · 2025-10-29

**Soundness:** 3
**Presentation:** 1
**Contribution:** 2
**Rating:** 2
**Confidence:** 5

**Summary:**

Concept-based models offer interpretability by predicting human-understandable concepts, but the prediction of the concepts themselves typically relies on black-box neural networks. This paper proposes a hierarchical concept prediction framework that learns an acyclic graph of concepts, aiming to improve interpretability.

**Strengths:**

1. The idea of hierarchically predicting concepts is sound and well-motivated.

**Weaknesses:**

**Major Weaknesses**

1. **Lack of Discussion on Related Work:**
There is a line of prior work that uses logic rules as model concepts [1, 2, 3]. The proposed approach seems similar in that a neural network selects rules based on their embeddings. However, the paper does not provide a clear discussion of how it differs from these methods.

2. **Concepts Are Not a True Bottleneck:**
A major concern is that in the proposed framework, concepts no longer act as a true information bottleneck. As shown in Figure 1, there appear to be direct black-box paths from input to output that may bypass the concept layer entirely. This could undermine the interpretability claim, as the model might “cheat” by encoding all information through embeddings rather than concept structure [2, 3].

3. **Methodology Unclear and Not Well-written:**
First, it is not explained how the concept pool is constructed. It is unclear whether the model relies on human-annotated concepts (as in datasets like CUB) or learns the concepts automatically. If the former, this raises concerns about scalability and generalizability; if the latter, it is unclear how the learned concepts maintain interpretability.
Second, the motivation behind parameterizing the encoder with a Bernoulli distribution and enforcing a delta distribution for the concept embeddings is not well justified. I suspect that this choice might have been made for training stability or sparsity, but it is not clear whether it plays a crucial role in performance. If this probabilistic formulation is important, it should be supported by ablation studies; if not, the reasoning for adopting it should be clarified.
Third, the description of the decoder mechanism - particularly how the model aggregates a pool of candidate rules based on parent concepts and selects one for child concepts - is confusing. From the current explanation, I suspect that the decoder may sequentially sample rules conditioned on parent availability, but this process is not clearly described. This lack of clarity makes it difficult to understand how the hierarchical structure is actually learned or enforced.
Finally, many notations and equations in Sections 2.2.2, 2.2.3, and 3 are not properly defined, which adds further confusion and makes the paper challenging to follow.

4. **Ambiguity in Likelihood Maximization (Eq. 9):**
Equation (9) suggests maximizing the likelihood of predicted concepts $\hat{c}_i$, implying access to their ground-truth labels. However, it is unclear whether such supervision is available or how the model learns the “correct” concepts if not.

5. **Non-Deterministic Interpretations:**
It appears that the decoder selects rules probabilistically, which raises the concern that the same input-output pair could yield different interpretations across runs.

6. **Questionable Use of "Theorem:**
The statements labeled as Theorem 5.1 and Theorem 5.2 do not appear to present substantial theoretical results. It is not evident what value these “theorems” contribute.

7. **Limited Scope and Evaluation:**
The experiments are restricted to simple vision datasets. Moreover, the paper later reveals that concept annotations are required, which significantly limits practicality. There is also no qualitative demonstration of interpretability, despite this being the paper’s main goal.

**Minor Comments**

8. In several equations, $\hat{x}$ appears where $x$ would be expected. Please clarify the distinction between the two.

**References**

[1] Deep Neural Networks Constrained by Decision Rules (AAAI 2019)

[2] Self-Explaining Deep Models with Logic Rule Reasoning (NeurIPS 2022)

[3] Toward Faithful and Human-Aligned Self-Explanation of Deep Models (npj Artificial Intelligence 2025)

**Questions:**

1. Clarify whether and how the model architecture enforces the concept bottleneck. Consider adding experimental evidence (e.g., ablation or information flow analysis) showing that the model genuinely relies on concepts rather than shortcutting through embeddings. (W2)

2. Provide a clearer, methodological explanation, including the presence of ground-truth labels for concepts and whether the interpretations are deterministic. (W3, W4, W5)

3. Consider rephrasing these sections as Propositions or Lemmas, or alternatively, move them to an appendix if they are primarily for clarification. Explicitly state what each result contributes to the method or guarantees about model behavior. (W6)

4. Evaluate on larger-scale datasets (e.g., ImageNet) to demonstrate scalability. Include at least one qualitative visualization or example of model interpretation. Discuss how the method could be extended to domains lacking concept annotations (e.g., via weak supervision or learned concepts). (W7)

5. Include a more detailed comparison with prior works that integrate logic reasoning into neural networks. Highlight what conceptual or methodological novelty your hierarchical approach introduces beyond embedding-based rule selection. (W1)

---

> ### Author Response · Authors · 2025-11-21
> **Rebuttal (Part 1)**
>
> We thank the reviewer for taking the time to read our paper.
>
> &nbsp;
>
> > There is a line of prior work that uses logic rules as model concepts [1, 2, 3], which seems similar in that a neural network selects rules based on their embeddings. However, the paper does not provide a clear discussion of how it differs from these methods. Can you highlight what conceptual or methodological novelty your hierarchical approach introduces beyond embedding-based rule selection?
>
> **While these are not works with supervised concepts, we will add them in our Related Work section. Because of the many differences, they do not undermine our novelty.** For instance:
> - **[1-3] build rules on input features, while H-CMR builds rules on concepts** that need to be predicted from the input features. This makes e.g. an experimental comparison impossible, as they do not have concept prediction and metrics like intervenability cannot be computed for these approaches.
> - **[1-3] have a flat model structure, while H-CMR has a hierarchical one.** This helps H-CMR with explaining concept predictions (which typical CBMs cannot do) and improves its intervenability.
> - **[1] assumes a set of decision rules is given, while [2, 3] and H-CMR learn rules entirely from scratch.** However, in [2, 3], the set of learned rules is implicit in the model's weights, similarly as for the concept-based model DCR [9], where during inference a neural network outputs a logic rule. **For [2, 3, 9], rule-based explanations are only local, while for H-CMR the entire set of learned rules is fully transparent and can be seen as global explanations of the model (Section 5.2).**
>
> &nbsp;
>
> > In the proposed framework, concepts are no longer a true information bottleneck. There appear to be direct black-box paths (through latent embeddings) from input to output that may bypass the concept layer entirely. This could undermine the interpretability claim, as the model might “cheat” by encoding all information through embeddings rather than concept structure [2, 3]. Can you clarify whether and how the model architecture enforces the concept bottleneck?
>
> **Using latent embeddings is the trend in current CBMs. This introduces some opacity, but this is a deliberate design choice**, allowing H-CMR to be as expressive as black-box neural networks. We follow a large line of CBM works which incorporate a latent embedding in the model (e.g. [1-6]), which sacrifices some interpretability in order to achieve black-box accuracy. However, in contrast to most such works (which forward the embedding through a neural network [1-5]), our embedding is used in a more interpretable way (i.e. it selects logic rules for inference). This way, **for us, this is less of a problem compared to other CBMs, as concepts still always form the final prediction** (through the logic rule execution), and cannot be bypassed using the embedding.
>
> **This is confirmed empirically by our high intervenability: the model uses the concepts extensively (Figure 5)** If the model would purely rely on the embedding, concept interventions would not affect accuracy at all.
>
> &nbsp;
>
> > Many notations and equations in Sections 2.2.2, 2.2.3, and 3 are not properly defined, which adds further confusion and makes the paper challenging to follow.
>
> We believe we explain all symbols and notations in the text. Could you mention which symbols or notations are insufficiently defined?
>
> &nbsp;
>
> > There is no qualitative demonstration of interpretability, despite this being the paper’s main goal. Can you include at least one qualitative visualization or example of model interpretation?
>
> **We provide examples of learned logic rules in Table 3 in the appendix.** H-CMR is locally interpretable because each prediction is made by evaluating such neurally selected logic rules (Section 5.2). On top of this, all the learned rules together form global explanations for the predictions.
>
> &nbsp;
>
> > What is the motivation behind parameterizing the encoder with a Bernoulli distribution and enforcing a delta distribution for the embeddings?
>
> **These are parametrizations that most CBMs and probabilistic Deep Learning models use.** Sigmoids are parametrizations of Bernoulli distributions, and Delta distributions are used to represent deterministic functions (e.g. hidden layers).

---

> ### Author Response · Authors · 2025-11-21
> **Rebuttal (Part 2)**
>
> > It is not explained how the concept pool is constructed. It is unclear whether the model relies on human-annotated concepts (as in datasets like CUB) or learns the concepts automatically. If the former, this raises concerns about scalability and generalizability; if the latter, it is unclear how the learned concepts maintain interpretability.
>
> Constructing the concept pool is a necessary condition for providing explanations, as the model should know what the language is it should provide explanations in. **For this, either the human defines the concepts (so they are part of the dataset) or a Vision-Language Model provides them** [10]. Our experiments investigate both datasets with concept labels (CUB, MNIST-Addition) and a dataset where we exploit a VLM (CIFAR-10). **This is a core methodological property of concept-based models.**
>
> &nbsp;
>
> > Equation (9) implies access to concept ground-truth labels. It is unclear whether such supervision is available. Also, doesn't concept supervision limit practicality? Can you discuss how the method could be extended to domains lacking concept annotations (e.g., via weak supervision or learned concepts).
>
> **As is standard in the CBM community, concepts are indeed explicitly supervised to align them with the human.** In practice, the dataset either comes with concept labels or vision-language models provide them [10]. In our experiments, we show H-CMR works with either of them (CUB and MNIST-Addition have concept labels, for CIFAR-10 we use a VLM). **While manually labelling the dataset limits practicality, the VLM-based approach only comes with limited additional compute.**
>
> **We have an experiment exploiting weak supervision** where we show that H-CMR can exploit existing expert knowledge to become significantly more data-efficient, operating well on MNIST-Addition with only 1% of labels (Figure 6). **Contrary to H-CMR, typical CBMs are unable to incorporate expert knowledge like this** (see "model interventions", e.g. in Table 1).
>
> &nbsp;
>
> > The description of the decoder mechanism - particularly how the model aggregates a pool of candidate rules based on parent concepts and selects one for child concepts - is confusing. I suspect that the decoder may sequentially sample rules conditioned on parent availability.
>
> **The decoder predicts each concept _exactly_ using Equation 2: the parent concepts are used for probabilistically selecting logic rules and evaluating them.** The rule selection and evaluation can be seen as a weighted average, where in the limit all probability mass goes to the evaluation of a single rule (i.e. categorical semantics). We compute this equation exactly, without any sampling.
>
> The model is hierarchical because of this parent-child relationship between concepts: **the decoder first predicts the source nodes of the learned concept graph, then their children, and so on.**
>
> &nbsp;
>
> > It appears that the decoder selects rules probabilistically, which raises the concern that the same input-output pair could yield different interpretations across runs.
>
> As mentioned above, we model uncertainty over the rule selection, but this probability is used in an exact way in practice (Equation 2). The model keeps trace of all rule evaluations, each weighted by their corresponding rule selection probability. This is for instance akin to probabilistic parses in grammars. There is no sampling involved, computations (and thus interpretations) are deterministic.
>
> &nbsp;
>
> > The experiments are restricted to simple vision datasets. Can you evaluate on larger-scale datasets (e.g. ImageNet) to demonstrate scalability?
>
> **We use datasets that are commonly used to benchmark CBMs (CUB, CIFAR-10, MNIST-Addition).** CUB is the most used CBM dataset which comes with human-provided concept labels. CIFAR-10 is a widely used dataset in Machine Learning that does not have any concept labels. **We show with CIFAR-10 that our method can use VLMs to supervise concepts on more general Deep Learning datasets _without any concept labels_.**
>
> While we can use VLMs to evaluate on larger-scale datasets without concept labels like ImageNet, ImageNet does not naturally come with any concepts.

---

> ### Author Response · Authors · 2025-11-21
> **Rebuttal (Part 3)**
>
> > What do Theorems 5.1 and 5.2 contribute? Either explicitly state what each result contributes to the method or guarantees about model behavior, or consider rephrasing them as Propositions or Lemmas, or move them to an appendix.
>
> These are important theoretical results regarding the expressivity of H-CMR.
> - Theorem 5.1 shows that H-CMR is as expressive as a neural network for classification irrespective of the chosen concept set, which is not the case for many CBMs. **This explains why H-CMR can achieve high accuracy irrespective of the concept set.**
> - Theorem 5.2 shows that H-CMR's parametrization of the learned concept graph enforces acyclicity (which a competitor like [11] does not) and is expressive enough to represent any directed acyclic graph. **Therefore, any dependency relation between concepts can be represented, which explains why H-CMR can achieve high intervenability.**
> **We will add these explanations below the theorems, and will instead call them propositions.**
>
> &nbsp;
>
> > In several equations, x_hat appears where x would be expected. Please clarify the distinction between the two.
>
> With x_hat, we mean X=x_hat, i.e. an assignment to a random variable X (footnote 2, page 4). Capital letters denote random variables. We do not use small x.
>
> &nbsp;
>
> [1] Deep Neural Networks Constrained by Decision Rules. Okajima et al.
>
> [2] Self-Explaining Deep Models with Logic Rule Reasoning. S Lee et al.
>
> [3] Toward Faithful and Human-Aligned Self-Explanation of Deep Models. S Lee et al.
>
> [4] Concept embedding models. Zarlenga et al.
>
> [5] Post-hoc concept bottleneck models. Yuksekgonul et al.
>
> [6] Concept bottleneck models with additional unsupervised concepts. Sawada et al.
>
> [7] Promises and pitfalls of black-box concept learning models. Mahinpei et al.
>
> [8] Incremental residual concept bottleneck models. Shang et al.
>
> [9] Interpretable neural-symbolic concept reasoning. Barbiero et al.
>
> [10] Label-free concept bottleneck models, Oikarinen et al.
>
> [11] Causal concept graph models: beyond causal opacity in deep learning. Dominici et al.

---

> > ### Comment · Reviewer_3RNr · 2025-11-26
> >
> > Thank you for the detailed rebuttal. While some of my initial concerns (e.g., regarding related work and the concept bottleneck) are now partially addressed, several important issues remain.
> >
> > **1. Notation and equations**
> > I still find the notation and mathematical presentation difficult to follow. For example: What is the precise distinction between $x$ and $\hat{x}$? What does $\hat{e}$ represent in Eq. (1)? What is $\hat{e}_{parents(i)}$ in Eq. (2)? These are just a few illustrations; similar issues appear throughout the paper. It may be difficult for reviewers to identify and list every ambiguous symbol individually in every equation, so I encourage the authors to conduct a thorough pass to ensure that all notation is clearly introduced and used consistently.
> >
> > **2. Qualitative demonstration of interpretability** I did check Table 3, but it offers only a very limited view of the learned rules. For a paper whose main goal is interpretability, I would expect a more complete qualitative example that shows, for a given input, (1) the original sample, (2) the corresponding explanation (e.g., instantiated rules or concepts), and (3) how this explanation aligns with the model’s prediction. A more explicit, end-to-end qualitative case study would make the interpretability claims much more convincing.
> >
> > **3. Bernoulli and delta distributions** Regarding the encoder, referring to a sigmoid output as ‘parameterizing a Bernoulli distribution’ is technically correct but not a commonly used wording and may confuse readers. More importantly, using the term “delta distribution” to describe selecting a single embedding from a pool is misleading. A Dirac delta distribution has a precise meaning in probability theory; if the method is not explicitly treating embeddings as random variables with such a distribution, this terminology is likely to cause confusion rather than clarify. I suggest either avoiding probabilistic terminology here or using it only where a genuinely probabilistic interpretation is essential and clearly justified.
> >
> > **4. Concept pool construction and supervision** From the rebuttal, it is now clear that the method requires either human-annotated concepts, or the use of a VLM to generate or supervise concepts. This is a significant methodological assumption and should be stated explicitly in the main paper (e.g., in the methodology section), not left implicit. Even if this is “standard in the CBM community,” reliance on human concepts or large VLMs has a real impact on practicality and scalability, and should be clearly discussed as a limitation and design trade-off.
> >
> > **5. Decoder mechanism and “probability”** From the explanation, my understanding is that the model computes weights over rules for each concept (conditioned on parent concepts), and uses these weights deterministically rather than via sampling.
> >
> > If these quantities are meant as soft selection weights that are always used deterministically, calling them “probabilities” may be misleading - especially when combined with earlier probabilistic terminology. It might be clearer to refer to them as just weights, unless you explicitly rely on a probabilistic interpretation throughout the framework.
> >
> > **6. Evaluation on larger-scale datasets** I appreciate that CUB, CIFAR-10, and MNIST-Addition are standard CBM benchmarks. However, recent concept-based and rule-based works - such as Concept Embedding Models, Deep Concept Reasoner, and Concept-based Memory Reasoner - have demonstrated results on larger-scale datasets (e.g., CelebA), which are substantially more challenging. Given this trend, simply stating that datasets like ImageNet lack concepts is less convincing, especially since VLMs are already used for supervision.
> >
> > **7. Propositions 5.1 and 5.2 (formerly Theorems)** Even with the additional explanation, I remain unconvinced that Propositions 5.1 and 5.2 warrant a prominent, standalone theoretical section. The expressivity results, as stated, largely formalize the intuitive idea that sufficiently many rules can approximate complex decision boundaries or DAG structures. This could likely be conveyed succinctly in prose or relegated to an appendix. Moreover, these propositions are not clearly tied back to concrete design choices or guarantees needed for the method to work in practice, nor are they referenced later in a way that reveals a strong conceptual dependency.
> >
> > **8. Clarification on $\hat{x}$** The rebuttal’s explanation that "with $\hat{x}$, we mean $X=\hat{x}$" unfortunately adds to the confusion. If $\hat{x}$ simply denotes a particular value of a random variable $X$, it is unclear why the notation $\hat{x}$ is needed instead of using $X$ consistently. More generally, many of the equations seem to mix random variables and specific instances in ways that are non-standard for mathematical notation in machine learning papers. This reinforces my earlier point that a careful, global clean-up of notation is necessary.

---

> ### Author Response · Authors · 2025-11-27
>
> > I still find the notation and mathematical presentation difficult to follow. For example:
> > - What is the distinction between X and x_hat?
> > - What does e_hat represent in Eq. (1)?
> > - What is c_hat_parents in Eq. (2)?
> >
> > These are just a few illustrations; similar issues appear throughout the paper. It may be difficult for reviewers to identify and list every ambiguous symbol individually in every equation, so I encourage the authors to conduct a thorough pass to ensure that all notation is clearly introduced and used consistently.
>
> - Capital letters denote random variables, and small letters with a hat denote assignments to these random variables (footnote 2). X is a random variable over images, x_hat is a concrete assignment to it (an image).
> - e_hat is predicted by the neural network g (Eq. 1) and is an embedding assigned to the random variable E (mentioned two lines below Eq. 1).
> - c_hat_parents are the assignments to the parent concepts random variables (mentioned in the line above Eq. 2, and also in the accolade below Eq. 2).
>
> **Each of these symbols is introduced unambiguously directly above/below the corresponding equation. If the reviewer considers other notations difficult to follow, could they clarify which notations?**
>
> &nbsp;
>
> > The rebuttal’s explanation that "with x_hat, we mean X = x_hat" adds to the confusion. If x_hat simply denotes a particular value of a random variable X, it is unclear why the notation x_hat is needed instead of using X consistently. More generally, many of the equations seem to mix random variables and specific instances.
>
> **These are distinct mathematical objects, so writing X in places where we mean a specific assignment x_hat would be incorrect.** This distinction between uppercase random variables and lowercase realizations is standard in probability theory and widely used in the machine learning literature.
>
> We cannot find any inconsistent use of the notation with respect to the aforementioned semantics. Could the reviewer refer to where they believe this is the case?
>
> &nbsp;
>
> > I did check Table 3, but it offers only a very limited view of the learned rules. For a paper whose main goal is interpretability, I would expect a more complete qualitative example that shows, for a given input, (1) the original sample, (2) the corresponding explanation (e.g., instantiated rules or concepts), and (3) how this explanation aligns with the model’s prediction.
>
> **Our interpretability claim follows directly from modelling each prediction as the evaluation of a logic rule using concepts.** In _explainability_, qualitative examples are common, but not in _interpretability_. **A logic rule is interpretable due to its semantics:** its operators (i.e. conjunction and negation) have a clear meaning to the human, and its atoms (i.e. concepts) are interpretable because they are explicitly supervised to be aligned to a human interpretation.
>
> &nbsp;
>
> > Regarding the encoder, referring to a sigmoid output as ‘parameterizing a Bernoulli distribution’ is technically correct but not a commonly used wording and may confuse readers. More importantly, using the term “delta distribution” to describe selecting a single embedding from a pool is misleading. A Dirac delta distribution has a precise meaning in probability theory; if the method is not explicitly treating embeddings as random variables with such a distribution, this terminology is likely to cause confusion rather than clarify. I suggest either avoiding probabilistic terminology here or using it only where a genuinely probabilistic interpretation is essential and clearly justified.
>
> **The encoder parametrizes a Bernoulli random variable (C) and a delta distribution (E) from our probabilistic graphical model.** We do not see how our description of them as such is misleading. Note that the delta distribution is used to represent embeddings (line 168), not the selection process (Equation 2, line 188).
>
> &nbsp;
>
> > From the explanation, my understanding is that the model computes weights over rules for each concept, and uses these weights deterministically rather than via sampling. If these quantities are meant as soft selection weights that are always used deterministically, calling them “probabilities” may be misleading - especially when combined with earlier probabilistic terminology. It might be clearer to refer to them as just weights, unless you explicitly rely on a probabilistic interpretation throughout the framework.
>
> **These quantities are not soft selection weights; they are probabilities of random variables.** Our explanation as a "weighted average" was for clarification. Probabilities does not mean one has to _sample_; many models can allow for **exact probabilistic inference** without sampling.

---

> > ### Author Response · Authors · 2025-11-27
> >
> > > From the rebuttal, it is now clear that the method requires either human-annotated concepts, or the use of a VLM to generate or supervise concepts. This is a significant methodological assumption and should be stated explicitly in the main paper (e.g., in the methodology section), not left implicit.
> >
> > We have added this in lines 128-129 of the revised version of the text.
> >
> > &nbsp;
> >
> > > I appreciate that CUB, CIFAR-10, and MNIST-Addition are standard CBM benchmarks. However, recent concept-based and rule-based works - such as Concept Embedding Models, Deep Concept Reasoner, and Concept-based Memory Reasoner - have demonstrated results on larger-scale datasets (e.g., CelebA), which are substantially more challenging. Given this trend, simply stating that datasets like ImageNet lack concepts is less convincing, especially since VLMs are already used for supervision.
> >
> > **The scalability/complexity concerns of the majority of CBM models are w.r.t. the number of concepts. CelebA is in this regard an easier dataset (40 concepts) than CUB (112 concepts)**: there are only a larger number of images (and supervisions) that makes concept prediction easier for CelebA. Moreover, the CelebA task is totally artificial, and we considered CUB a more sensible dataset to showcase a larger scale benchmark.
> >
> > &nbsp;
> >
> > > Even with the additional explanation, I remain unconvinced that Propositions 5.1 and 5.2 warrant a prominent, standalone theoretical section. The expressivity results, as stated, largely formalize the intuitive idea that sufficiently many rules can approximate complex decision boundaries or DAG structures. This could likely be conveyed succinctly in prose or relegated to an appendix. Moreover, these propositions are not clearly tied back to concrete design choices or guarantees needed for the method to work in practice, nor are they referenced later in a way that reveals a strong conceptual dependency.
> >
> > **We believe (and agree with reviewer 99og) that this is an important result**, not to be relegated to an appendix. It clearly shows that **there is no _theoretical_ expressivity advantage in using a pure black-box model w.r.t. to our model**. This is often considered a downside of neurosymbolic (and concept-based) models, where their expressivity is limited by the number (and availability) of rules (and concepts). Our model expressivity does not suffer from any of these two sources.

---

### Official Review · Reviewer_U5fB · 2025-11-02

**Soundness:** 3
**Presentation:** 3
**Contribution:** 3
**Rating:** 2
**Confidence:** 4

**Summary:**

This paper introduces Hierarchical Concept Memory Reasoner (H-CMR), a new Concept-Based Model (CBM) that provides interpretability not only for the final task but also for intermediate concept predictions, which learns a DAG of concepts and represents their relationships via neural-selected symbolic logic rules stored in a shared memory. Specifically, H-CMR contains three different modules: an encoder which predicts source concepts and a latent embedding, a decoder which hierarchically infers other concepts through symbolic rule execution, and a memory which stores learnable logic rules defining parent–child relations. Experiments show that H-CMR achieves SOTA concept accuracy and maintains universal classification capability comparable to black-box networks.

**Strengths:**

1. The idea is interesting. Compared to previous CBM methods, which can only explain task-level predictions, H-CMR explicitly models how concepts depend on one another via interpretable logic rules.
2. The paper is well-written and easy to follow.
3. The experiment results on three different datasets show the effectiveness of the proposed methods.

**Weaknesses:**

1. Although the idea is interesting. However, I feel suspicious about whether the proposed methods can be extended to real-world scenarios. Currently, the experiments are done on some small toy datasets, which limit the contribution of the proposed methods. In order to provide a more comprehensive view of the proposed methods, it is better to show the results of the model in more complex scenarios. For example, there are lots of different vehicle categories in ImageNet. Is it possible to use the H-CRM to extract concepts that humans can understand and explain the logic between different concepts and tell the logic between different concepts? If humans cannot understand the extracted concept, the interpretability is not valid, even if humans can understand the logic between concepts. For example, a model can tell me $C_k=C_i \vee C_j$, without any understanding of $C_k, C_i, C_j$, we can not even tell whether the logic is true or false. Also, without such understanding, model intervention is not valid because we cannot even tell how to change the logic behind the concepts.
2. The ablation studies are not sufficient. Currently, the H-CMR claims to provide interpretable inference for both concept and task predictions, the paper does not include any empirical validation that the extracted explanations (logic rules or hierarchical DAGs) are semantically or causally correct.

**Questions:**

I will raise my rating if the author can provide additional evidence of the interpretability and demonstrate that it is correct.

---

> ### Author Response · Authors · 2025-11-21
> **Rebuttal**
>
> We want to thank the reviewer for taking the time to read our paper. We are pleased the reviewer considers the idea interesting, the paper well-written and our experiments and their results effective.
>
> &nbsp;
>
> > Can the proposed methods can be extended to real-world scenarios, e.g. using ImageNet? Can H-CRM extract concepts that humans can understand and explain the logic between different concepts? If humans cannot understand the extracted concept, the interpretability is not valid, even if humans can understand the logic between concepts. For example, a model can tell me C_k = C_i or C_j, but without any understanding of C_k, C_i, C_j, we cannot even tell whether the logic is true or false.
>
> We agree with the reviewer: in any logic language, semantics come from the joint interpretation of the symbols (concepts) and the operators. **In order to provide semantics for the concepts, we directly supervise them.** This is a standard practice in CBMs to avoid misalignment. In CBM works, either the datasets have concept labels, or a vision-language model is used to supervise the concepts. In our experiments we do both (CUB, MNIST-Addition and Synth has concept labels, for CIFAR-10 we use a VLM).
>
> In order to understand predictions, the user should provide a set of concepts to the model, which the explanation should be based on. Unfortunately, no concept set is available in ImageNet.
>
> &nbsp;
>
> > H-CMR claims to provide interpretable inference for both concept and task predictions, but the paper does not include any empirical validation that the extracted explanations (logic rules or hierarchical DAGs) are semantically or causally correct.
>
> In interpretability, we want to understand the causal reasoning of the model without necessarily worrying about whether it catches the semantically valid or causally correct relations in the real world. Of course, if they don't match, this could be problematic, but still the method will be able to show this.
>
> For this reason, **it is transparent in our model _how the model predicts concepts and tasks using each other_, which is a direct consequence of using logic rules**. We aim to make transparent to the human how the model reasons, whether it is wrong or right (lines 90-91, Section 5.2). If the human disagrees with the reasoning process, they may do model interventions to change it. **This is the interpretability that H-CMR provides.**
>
> In Table 3 in the appendix, we still provide examples of learned rules to show they generally make sense to the human. We can move this table to the main text.

---

> ### Author Response · Authors · 2025-11-28
>
> Dear reviewer,
>
> Please let us know if you have any further questions or things we could clarify further. If not, we would appreciate if you could consider updating your review based on our replies.

---

### Official Review · Reviewer_99og · 2025-11-03

**Soundness:** 3
**Presentation:** 3
**Contribution:** 3
**Rating:** 6
**Confidence:** 3

**Summary:**

The authors proposed H-CMR (Hierarchical Concept Memory Reasoner), a neuro-symbolic framework that performs interpretable concept reasoning through a directed acyclic graph of learned logical rules.
It first predicts semantically grounded concepts from perceptual inputs and then composes these concepts hierarchically via differentiable rule modules to infer higher-level concepts and final task decisions.
Each rule defines an explicit logical relation among parent concepts, enabling transparent reasoning across multiple abstraction levels.
The model jointly optimizes concept prediction, rule learning, and task objectives to balance interpretability with performance.
Experimentally, H-CMR is validated across synthetic and visual benchmarks such as MNIST-Addition, CIFAR-10, and CUB, demonstrating improved intervenability and interpretability over concept-based baselines.
These results highlight H-CMR’s ability to achieve structured, human-understandable reasoning without sacrificing predictive accuracy.

**Strengths:**

- **S1. Strong integration of concept-based and neuro-symbolic reasoning**

The proposed method provides a well-structured integration of concept bottleneck modeling with neuro-symbolic reasoning through its unified architecture of concept learning and rule-based inference.
It jointly learns concept embeddings and logical rules within a directed acyclic graph, allowing the model to represent both the semantics and dependencies among concepts in an interpretable manner.
The neural encoder captures perceptual information and grounds the concepts, while the symbolic reasoning layer operates over these learned concepts using explicit logical compositions. This design effectively connects data-driven neural representations with transparent symbolic reasoning, enabling interpretable predictions at both the concept and task levels.

- **S2. Clear formulation of hierarchical reasoning and strong theoretical grounding**

H-CMR clearly defines how hierarchical reasoning emerges through the recursive composition of learned logical rules.
Each concept is inferred from its parent concepts via explicit rule evaluation, and these dependencies propagate across multiple levels of the directed acyclic graph (DAG), forming interpretable hierarchical structures.
Also, the authors provided solid theoretical grounding for this formulation.
It formalizes the model’s expressivity, acyclicity, and computational complexity, with clear statements of the underlying assumptions and guarantees.

**Weaknesses:**

- **W1. Computational complexity and scalability considerations**

While the proposed method offers a principled formulation for interpretable concept reasoning, its computational complexity raises concerns about practical scalability.
Empirical results in Table 4 show that runtime and memory usage increase significantly with the number of concepts, suggesting that efficiency could become a bottleneck for large-scale or densely connected concept graphs.
It would be valuable to explore how the practicability of the proposed method could be improved--particularly by testing it on real-world, higher-dimensional datasets or by incorporating optimization strategies such as hierarchical pruning, rule caching, or sparse evaluation. Such extensions could strengthen the framework’s applicability beyond medium-scale benchmarks while preserving its interpretability advantages.

- **W2. Need for stronger qualitative concept visualization and complementary quantitative validation**

While the proposed method provides clear rule-based reasoning and interpretable dependency graphs, it would benefit from richer qualitative visualization of the learned concepts--a practice that has become standard in concept-based interpretability research. Many concept bottleneck and concept-based models support their claims with visual examples showing how individual concepts are localized, clustered, or activated across samples. Such qualitative results help readers intuitively grasp what each learned concept represents and how concept combinations contribute to hierarchical reasoning.
Incorporating visual analyses of representative concepts or rule activations--such as heatmaps, or example patches--would make the interpretability of H-CMR more tangible and relatable.
In parallel, pairing these visuals with quantitative measures (e.g., localization accuracy, sparsity, or activation consistency across samples) could provide stronger empirical support for the model’s interpretability claims.

This also opens several natural questions for further exploration:
- How can concept localization be structured hierarchically, reflecting parent–child relationships in the rule graph?
- Can hierarchical visualizations reveal whether higher-level concepts emerge from spatial or semantic composition of lower-level ones?
- How can visual concept maps be used to validate the internal reasoning paths of H-CMR?

Addressing these questions through both visualization and quantitative validation would strengthen the link between H-CMR’s symbolic reasoning and the perceptual evidence that grounds its concepts, making the framework more convincing and complete.

**Questions:**

Most of my main concerns or questions have been outlined in the Weaknesses section.

---

> ### Author Response · Authors · 2025-11-21
> **Rebuttal**
>
> We want to thank the reviewer for taking the time to read our paper. We are pleased the reviewer considers our work a strong integration of concept-based and neuro-symbolic reasoning with a clear formulation of hierarchical reasoning with a strong theoretical grounding.
>
> &nbsp;
>
> > The computational complexity raises concerns about practical scalability. Empirical results in Table 4 show that runtime increase significantly with the number of concepts.
>
> **Table 3's total training time includes the time for training the competitors, which take significantly more time than H-CMR.** For the results in Figure 4, the total training time was 2.7h for H-CMR, 4.4h for SCBM and 17.9h for CGM.
>
> &nbsp;
>
> > It would be valuable to explore how the practicability (computational complexity) of the proposed method could be improved – particularly by testing it on real-world, higher-dimensional datasets or by incorporating optimization strategies such as hierarchical pruning, rule caching, or sparse evaluation.
>
> **H-CMR has the same worst-case computational complexity as the state-of-the-art.** This is quadratic in the number of concepts, exactly like the main competitors CGM and SCBM. It also has built-in ways to reduce the complexity for both training and inference. During training, this complexity can be reduced by doing model interventions, e.g. by forcing some edges to be absent. During inference, this quadratic complexity is worst-case, with the actual one depending on how many concepts are used in the learned rules. H-CMR has a prototype regularization scheme (line 858) with a hyperparameter that can be used to tune rule sparsity. This means the human can reduce the computational complexity by tuning this hyperparameter (or by doing model interventions).
>
> &nbsp;
>
> > The proposed method provides clear rule-based reasoning and interpretable dependency graphs, but it would benefit from richer qualitative visualization of the learned concepts. You could show how individual concepts are localized, clustered, or activated across samples. Incorporating visual analyses of representative concepts or rule activations--such as heatmaps, or example patches--would make the interpretability of H-CMR more tangible and relatable.
>
> We believe there are two main components to this: (1) visualising the learned concepts (e.g. using saliency maps) and (2) visualising the rules.
>
> **Like most concept-based works, we do not focus on visualising learned concepts.** Such analyses are common in works that do _concept discovery_, which we don't do. For us, like many other concept-based works, concepts are explicitly supervised in order to be aligned with the human. A smaller number of these also visualise the concepts, but this is typically because they focus more on the low-level perceptive component, e.g. learning localized concepts, which is orthogonal to what most works (like ours) do.
>
> **For visualising the rules, we provide examples of learned rules in Table 3** in the appendix to show the learned rules are intuitive to humans.
>
> **We also want to remark that H-CMR is automatically locally interpretable because the final step in each prediction consists of evaluating logic rules** (Section 5.2). For instance, this interpretability enables the human to perform _model interventions_ to change the rules if the human does not agree with them (e.g. if they are biased) (Section 5.3).

---

> ### Author Response · Authors · 2025-11-28
>
> Dear reviewer,
>
> Please let us know if you have any further questions or things we could clarify further. If not, we would appreciate if you could consider updating your review based on our replies.

---

### Official Review · Reviewer_zruZ · 2025-11-05

**Soundness:** 3
**Presentation:** 2
**Contribution:** 3
**Rating:** 4
**Confidence:** 3

**Summary:**

This paper introduces Hierarchical Concept Memory Reasoner (H-CMR), a concept-based model that combines symbolic reasoning with neural attention for interpretable concept and task prediction. The key view is to learn a directed acyclic graph (DAG) over concepts and tasks, where each node is predicted using neurally selected, symbolic logic rules. This framework enables step-by-step symbolic inference and supports both concept-level and model-level interventions.

**Strengths:**

1. More interpretability: The model offers clear, logic-based explanations for both concept and task predictions, going beyond typical CBMs that only explain task-level outputs.

2. Human-in-the-loop friendly: The architecture is explicitly designed to allow interventions during both inference and training, making it suitable for scenarios where expert knowledge is available.

**Weaknesses:**

1. Latent embeddings still required: Despite the symbolic reasoning layer, a latent embedding is still used in rule selection, which introduces some opacity.

2. Limited task diversity: Most experiments focus on standard classification datasets; it remains unclear how well the method generalizes to more complex domains.

3. Quadratic complexity: The approach has worst-case quadratic time complexity in the number of concepts, which may limit scalability.

**Questions:**

See above.

---

> ### Author Response · Authors · 2025-11-21
> **Rebuttal**
>
> We want to thank the reviewer for taking the time to read our paper, and we are pleased that the reviewer considers the types of explanations that H-CMR provides clear and acknowledges that they go beyond typical CBMs.
>
> &nbsp;
>
> > Despite the symbolic reasoning layer, a latent embedding is still used in rule selection, which introduces some opacity.
>
> **Using latent embeddings is the trend in current CBMs. We agree that this introduces some opacity, but this is a deliberate design choice**, allowing H-CMR to be as expressive as black-box neural networks. We follow a large line of CBM works which incorporate a latent embedding in the model (e.g. [1-6]), which sacrifices some interpretability in order to achieve black-box accuracy. However, in contrast to most such works (which forward the embedding through a neural network [1-5]), our embedding is used in a more interpretable way (i.e. it selects logic rules for inference). This way, **for us, this is less of a problem compared to other CBMs**, as concepts still always form the final prediction (through the logic rule execution), and cannot be bypassed using the embedding.
>
> **This is confirmed empirically by our high intervenability: the model uses the concepts extensively (Figure 5).** If the model would purely rely on the embedding, concept interventions would not affect accuracy at all.
>
> &nbsp;
>
> > Most experiments focus on standard classification datasets; it remains unclear how well the method generalizes to more complex domains.
>
> **We use datasets that are commonly used to benchmark CBMs (CUB, CIFAR-10, MNIST-Addition).** CUB is the most used CBM dataset which comes with human-provided concept labels. CIFAR-10 is a widely used dataset in Machine Learning that does not have any concept labels. **We show with CIFAR-10 that our method can use VLMs to supervise concepts on more general Deep Learning datasets without _any concept labels_.**
>
> &nbsp;
>
> > The approach has worst-case quadratic time complexity in the number of concepts, which may limit scalability.
>
> **H-CMR has the same worst-case computational complexity as the state-of-the-art.** This is quadratic in the number of concepts, exactly like the main competitors CGM and SCBM. While this complexity is a limitation for very large concept sets, we believe that it is acceptable in many practical CBM settings, where the number of concepts is often kept moderate to maintain interpretability. Moreover, the actual runtime (and complexity) depends on the learned rules, which can be sparsified via the prototype regularization hyperparameter (line 858).
>
> &nbsp;
>
> [1] Concept embedding models. Zarlenga et al.
>
> [2] Post-hoc concept bottleneck models. Yuksekgonul et al.
>
> [3] Concept bottleneck models with additional unsupervised concepts. Sawada et al.
>
> [4] Promises and pitfalls of black-box concept learning models. Mahinpei et al.
>
> [5] Incremental residual concept bottleneck models. Shang et al.
>
> [6] Interpretable neural-symbolic concept reasoning. Barbiero et al.

---

> ### Author Response · Authors · 2025-11-28
>
> Dear reviewer,
>
> Please let us know if you have any further questions or things we could clarify further. If not, we would appreciate if you could consider updating your review based on our replies.

---

### Author Response · Authors · 2025-12-03
**Summary for AC**

Dear AC,

We want to give a concise summary of the reviews, our rebuttal and our corresponding changes. We want to remark that all reviewers (except 3RNr) had not yet been able to respond to our rebuttal.


### Common points

- *Reviewer zruZ stressed our **worst-case complexity** (quadratic in the concepts), and Reviewer 99og was interested in ways to improve it.* We remarked that (1) our complexity is the **same as our competitors** and (2) our method has **already ways of tuning it**.
- *Reviewers zruZ and 3RNr considered our use of **a latent embedding** a downside.* We remarked that this is the **current trend in CBMs**, allowing us to achieve the same **expressivity** as neural networks. We also show (e.g. empirically) that it is for us **less of a problem than for other CBMs**.

### Reviewer zruZ

The reviewer was concerned about computational complexity, which is the same for our competitors, and the use of a latent embedding, which we contextualized within the literature.

- *They also mentioned we **only use standard classification datasets**.* We remarked that we show experimentally that our method can work on general classification datasets (also e.g. without concept labels).

### Reviewr 99og

The reviewer was concerned about runtime, but this was due to a misunderstanding when reading a table. The reviewer would also like qualitative visualisations, and we pointed out we already have some.

- *They mentioned concerns about **runtime after looking at Table 3**.* We remarked that **Table 3 denotes total runtime** for our experiments, with most of it spent on competitors.
- _They said the paper would benefit from **qualitative visualisations** of learned concepts and rules._ We remarked that we provide **examples of learned rules** in our appendix, and that we do not provide visualisations of concepts, as our concepts are supervised to align them to the human (like in most CBMs).

### Reviewer U5fB

The reviewer had a concern that we clarified by referring to an explicit, common assumption in our setup, and a concern about causal and semantic correctness, which falls outside the scope of our goals.

- *They asked whether the method could learn **meaningful concepts in real-world scenarios**.* We explained that we provide meaningful semantics to the concepts **by explicitly supervising them**, which is standard in concept-based works.
- *They mentioned we do not include an **empirical validation** that the **learned rules** and graphs are **semantically** or **causally correct**.* We remarked that learning this is **not our goal** (as mentioned in the text). We provide **interpretability**: the human may (1) inspect the rules and graphs, (2) (dis)agree with them, and (3) change them if desired.

### Reviewer 3RNr

The reviewer appeared to have limited familiarity with the standard terminology used in interpretable modeling and probabilistic deep learning. For instance, they questioned the use of a sigmoid as a parameterization of a Bernoulli distribution, despite our model being clearly described within a Bayesian Network framework. We have provided additional clarification in the review to address these points. However, we were somewhat surprised by the nature of these concerns, as they suggest a possible mismatch between the reviewer’s expertise and the subject matter of the paper.

- *They believed **notations and equations are undefined** and gave examples of undefined symbols.* We believe all symbols are explained in the text. In all the reviewer's examples, the **symbols are explained directly above**/below the equation.
- *They asked for **motivating the use of Bernoulli and Delta distributions**.* We explained those are common in probabilistic deep learning.
- *They found it unclear whether concept supervision is available, and asked whether our models can work with **weak supervision**.* We confirmed this and remarked that we **already have an experiment** with weak supervision.
- *They raised concerns that the **same input-output pair** could yield **different interpretations** due to having a probabilistic model.* We remarked that **this cannot happen** because we do not sample.
- *They believed **important related work was unreferenced**.* We explained why these works are **not closely related** and added them in our related work, stating the differences.
- *They mentioned there is **no qualitative demonstration of interpretability**.* We explained that predictions are **automatically interpretable** because the final step of the prediction is a **logic rule** evaluation.
- *They asked for evaluating on **larger-scale datasets** (e.g. ImageNet, CelebA) to show scalability.* We remarked that we use common datasets for benchmarking CBMs (CUB, CIFAR-10, MNIST-Add), and that **CUB has more concepts than CelebA**, thus being more difficult to scale to.
- *They believed our theorems do not warrant a theoretical section.* We stressed their importance (agreeing with Reviewer 99og).

---

### Meta-Review · Area_Chair_Y8Ha · 2026-01-06

**Summary:**

By carefully reviewing the comments from reviewers, I agree with their assessment that several critical issues prevent the paper from meeting the acceptance bar. In particular, concerns regarding clarity of presentation, limited empirical support for the interpretability claims, and the restricted scope of evaluation remain substantial. Although the rebuttal provided helpful conceptual clarifications and addressed some misunderstandings, it did not sufficiently resolve these core concerns. Given the high standards of ICLR as a top-tier conference, I therefore concur with the reviewers that the submission is not yet ready for acceptance, and my final recommendation is rejection.

**Reviewer Concerns:**

The rebuttal helped clarify several misunderstandings regarding the method’s formulation and assumptions, but the core concerns raised by multiple reviewers—particularly those related to presentation clarity, scalability, and the limited evaluation scope—remain not fully addressed.

**Reviewer Scores:**

The initial reviewer scores are 4/6/2/2, with three reviews being negative and one positive. The negative comments were dominant, and the concerns they raised were substantial. While the authors attempted to address these issues in the rebuttal, I believe it would be very difficult to overturn the overall negative assessment  or shift the consensus toward positive scores.

---

### Decision · Program_Chairs · 2026-01-26

Reject